



# Principal Component Gradiometer technique for removal of spacecraft-generated disturbances from magnetic field data

Ovidiu Dragoş Constantinescu[1,2], Hans-Ulrich Auster[1], Magda Delva[3], Olaf Hillenmaier[4], Werner Magnes[3], and Ferdinand Plaschke[3]

[1]Institute for Geophysics and Extraterrestrial Physics, TU Braunschweig, Germany
[2]Institute for Space Sciences, Bucharest, Romania
[3]Space Research Institute, Austrian Academy of Sciences, Graz, Austria
[4]Magson GmbH, Berlin, Germany

**Correspondence:** D. Constantinescu (d.constantinescu@tu-bs.de)

**Abstract.**

In situ measurement of the magnetic field using space borne instruments requires either a magnetically clean platform and/or a very long boom for accommodating magnetometer sensors at a large distance from the spacecraft body. This significantly drives up the costs and time required to build a spacecraft. Here we present an alternative sensor configuration and an algorithm allowing for ulterior removal of the spacecraft generated disturbances from the magnetic field measurements, thus lessening the need for a magnetic cleanliness program and allowing for shorter boom length. The proposed algorithm is applied to the Service Oriented Spacecraft Magnetometer (SOSMAG) onboard the Korean geostationary satellite GeoKompsat-2A (GK2A) which uses for the first time a multi-sensor configuration for onboard data cleaning. The successful elimination of disturbances originating from several sources validates the proposed cleaning technique.

## 1 Introduction

Since very early in space exploration it has become clear that the main limitation in performing accurate magnetic field measurements came not from the instruments themselves but rather from the strong artificial magnetic fields generated by the spacecraft carrying them. It was recognized that there are three possible approaches to mitigate this problem: One could limit the electromagnetic emissions coming from the spacecraft by going through a rigorous magnetic cleaning procedure. This is a costly and complicated engineering task and introduces limitations on building and operating other onboard instruments, see e.g. (Narvaez, 2004) for details on the magnetic cleanliness program for Cassini magnetic field experiment (Dougherty et al., 2004). Another approach is to accommodate the magnetometer at a large distance from the spacecraft, usually at the end of a long boom, such as the 12 m long Kaguya boom (Kato et al., 2010) or the 13 m long Voyager boom (Behannon et al., 1977). This introduces constrains on the spacecraft operations and still requires a certain degree of magnetic cleanliness of the spacecraft in order to keep the boom at reasonable length. A third way is to accept the presence of spacecraft generated disturbances in the measured magnetic field and to remove the artificial contributions afterwards onboard or on ground through special techniques (Mehlem, 1978; Georgescu et al., 2008; Pope et al., 2011). An extreme case, where no magnetic cleanliness and no




boom was provided is e.g. the magnetic field experiment on the MASCOT lander (Herčík et al., 2017). In most cases however, a combination of two or all of the approaches above is employed. For instance, Cluster (Escoubet et al., 1997) and THEMIS (Angelopoulos, 2008) are magnetically clean spacecraft carrying magnetometers on relatively long booms. For normal science investigations, the stray magnetic field from these spacecraft is well below the required accuracy and no further steps to remove it are usually necessary. Venus Express (Titov et al., 2006) on the other hand, was a magnetically dirty spacecraft with two magnetometers (Zhang et al., 2006) mounted on a short boom for which extensive data cleaning efforts had to be undertaken (Pope et al., 2011). A comprehensive overview of the instrumentation and challenges related to measuring magnetic fields in space is given by Balogh (2010). In this work we focus on the third approach: removal of the contribution of the spacecraft generated magnetic field from the measured data.

One of the first studies on using multi sensor measurements to clean magnetic field data measured onboard spacecraft, came from Ness et al. (1971). The proposed method was then successfully applied in a simplified manner to Mariner 10 magnetic field data (Ness et al., 1974) assuming one single dipole disturber source. Neubauer (1975) gave a detailed error analysis of the Ness et al. (1971) method and discussed optimum placement of collinear sensors. The more recent cleaning procedure used by Pope et al. (2011) for Venus Express, though based on the same principle, is much more sophisticated allowing removal of disturbances from several different sources. However, additional information about the spacecraft operation and fuzzy logic had to be used to distinguish between the disturbance sources. Such a complex algorithm would be difficult to implement for onboard data cleaning. Our aim is a correction method which reduces to a linear (or at most quadratic) combination of the the magnetic field values measured by several sensors without input from other sources, therefore easy to implement onboard.

Similarly with Ness et al. (1971) and Pope et al. (2011) methods, the disturbance removal method described in the following sections is based on the fact that the magnetic field measured by each sensor is the sum of the ambient magnetic field and the artificial magnetic field generated by the spacecraft. Because the ambient field is the same for all sensors, it vanishes in the difference between the measurements from any two sensors, similar with the gradiometer working principle. The difference is entirely determined by the artificial magnetic field sources from the spacecraft, preserving their time dependence. Magnetic disturbances generated by time-dependent currents flowing through simple mechanically fixed current loops keep constant direction, therefore in general the disturbance affects only one component of the measured field. In many cases the direction of the strongest disturbance coincides with the principal component of the measured field, allowing application of the correction only to the affected component.

The proposed method is applied to the SOSMAG instrument (Auster et al., 2016; Magnes et al., 2020) which, together with the Particle Detector experiment (Seon et al., 2020) is part of the Korea Space wEather Monitor (KSEM) (Oh et al., 2018) onboard the GeoKompsat-2A (GK2A) geostationary spacecraft. SOSMAG consists of four three-axial magnetic field sensors, two of them mounted on a short boom extended from the spacecraft, the other two placed near strong magnetic disturbance sources within the spacecraft. Once the correction coefficients are determined on ground, they are uploaded to the spacecraft and are used by the onboard software to correct in-flight the magnetic field measurements. This enables accurate magnetic field measurements onboard GK2A without the need of passing through a magnetic cleanliness program before launch.



The remaining of the paper is organized as follows: In sec. 2 we discuss the gradiometer principle on which our method is based. Sec. 3 outlines the proposed Principal Component Gradiometer (PiCoG) method to remove spacecraft generated disturbances from the measured magnetic field data. Sec. 4 describes how the PiCoG method is applied to clean the GK2A SOSMAG data. The limitations of the proposed method are discussed in sec. 5. Sec. 6 summarises our work.

## 2  Disturbances from known sources

This section gives the analytical expressions for disturbances when the exact locations of the magnetic field sources and of the sensors are known. While in most cases the direct application of these expressions is not practical, the section outlines the general principle used by gradiometer-based disturbance cleaning methods, namely the possibility to express the spacecraft generated disturbances in terms of differences between measurements taken at distinct places. The relations derived here constitute the basis of the PiCoG algorithm detailed in section 3. Because higher multipole moments attenuate strongly with the distance to the source and become negligible even for short booms, we will concentrate only on the dipole and quadrupole contributions.

### 2.1  Single disturbance source

The magnetic field produced at the position $\boldsymbol{r} = r\hat{\boldsymbol{r}}$ by a dipole characterised by a slowly varying time dependent magnetic moment $\boldsymbol{M}(t)$ is given by:

$$\boldsymbol{b}(\boldsymbol{r},t) = \frac{\mu_0}{4\pi r^3}\big(3\mathcal{X}(\hat{\boldsymbol{r}}) - \mathcal{I}\big)\boldsymbol{M}(t) \tag{1}$$

where the elements $X_{kl} = \hat{r}_k \hat{r}_l$ of the matrix $\mathcal{X}$ are given by the product between the components of the position versor $\hat{r}$, and $\mathcal{I}$ is the $3\times3$ identity matrix. Knowing the magnetic field at the position $\boldsymbol{r}^i$, one can compute the magnetic field at any position $\boldsymbol{r}^j$ without knowledge about the source magnetic moment $\boldsymbol{M}(t)$:

$$\boldsymbol{b}(\boldsymbol{r}^j,t) = \mathcal{T}^{\mathrm{dip}}(\boldsymbol{r}^i,\boldsymbol{r}^j)\boldsymbol{b}(\boldsymbol{r}^i,t) \tag{2}$$

where the time-independent linear transformation $\mathcal{T}^{\mathrm{dip}}$ is:

$$\mathcal{T}^{\mathrm{dip}}(\boldsymbol{r}^i,\boldsymbol{r}^j) = \left(\frac{r^i}{r^j}\right)^3 \big(3\mathcal{X}^j - \mathcal{I}\big)\big(3\mathcal{X}^i - \mathcal{I}\big)^{-1} \tag{3}$$

The inverse $(3\mathcal{X} - \mathcal{I})^{-1}$ always exists and is equal to $(^3\!/\!_2\mathcal{X} - \mathcal{I})$.

Similar relations can be written for a time-dependent quadrupole defined by its moment $\mathcal{Q}(t)$:

$$\boldsymbol{b}(\boldsymbol{r},t) = \frac{\mu_0}{4\pi r^4}\big(5\mathcal{X}(\hat{\boldsymbol{r}}) - 2\mathcal{I}\big)\mathcal{Q}(t)\hat{\boldsymbol{r}} \tag{4}$$

$$\boldsymbol{b}(\boldsymbol{r}^j,t) = \mathcal{T}^{\mathrm{quad}}(\boldsymbol{r}^i,\boldsymbol{r}^j)\boldsymbol{b}(\boldsymbol{r}^i,t) \tag{5}$$

$$\mathcal{T}^{\mathrm{quad}}(\boldsymbol{r}^i,\boldsymbol{r}^j) = \left(\frac{r^i}{r^j}\right)^4 \big(5\mathcal{X}^j - 2\mathcal{I}\big)\mathcal{G}^{ji}\big(5\mathcal{X}^i - 2\mathcal{I}\big)^{-1} \tag{6}$$





where $\mathcal{G}^{ji}$ is the rotation matrix which transforms the versor $\hat{r}^j$ to the versor $\hat{r}^i$ and $(5\mathcal{X} - 2\mathcal{I})^{-1}$ is equal to $(5/6\mathcal{X} - 1/2\mathcal{I})$.

To derive the above relations we used the fact that $\mathcal{Q}$ is a symmetric matrix, $\mathcal{Q}^T = \mathcal{Q}$ and $\mathcal{X}$ is an idempotent matrix, $\mathcal{X}^2 = \mathcal{X}$.

Assuming that the ambient magnetic field is generated by distant sources and thus it is the same at the positions $\boldsymbol{r}^i$ and $\boldsymbol{r}^j$, it is possible to separate the contribution $\boldsymbol{b}(\boldsymbol{r}^i, t)$ from a nearby dipole or quadrupole from the ambient field by computing the difference between the magnetic field at the two positions:

$$\boldsymbol{b}(\boldsymbol{r}^i, t) = \left(\mathcal{T}(\boldsymbol{r}^i, \boldsymbol{r}^j) - \mathcal{I}\right)^{-1} \left(\boldsymbol{B}_{\text{measured}}(\boldsymbol{r}^j, t) - \boldsymbol{B}_{\text{measured}}(\boldsymbol{r}^i, t)\right) \tag{7}$$

where the total measured magnetic field $\boldsymbol{B}_{\text{measured}}(\boldsymbol{r}, t) = \boldsymbol{B}(t) + \boldsymbol{b}(\boldsymbol{r}, t)$ contains both the position-independent ambient magnetic field $\boldsymbol{B}(t)$ and the position-dependent disturbance magnetic field $\boldsymbol{b}(\boldsymbol{r}, t)$. Sensor specific disturbances such as sensor noise and sensor offset will be considered later.

Note that the $\mathcal{T}$ matrices only depend on the position vectors $\boldsymbol{r}^i$ and $\boldsymbol{r}^j$. They are independent on the dipole $\boldsymbol{M}(t)$ or quadrupole $\mathcal{Q}(t)$ moments and perform a similar function with the propagator operator in quantum mechanics. Equation (7) shows that once the $\mathcal{T}$ matrix is determined for a pair of sensors, measurements from those two points are sufficient to separate the contribution from a single magnetic field source with arbitrary time variation from the ambient magnetic field.

## 2.2 Multiple disturbance sources

The contributions from more than one simultaneously active, arbitrary placed source with arbitrary time dependence cannot be separated from the ambient field in a simple way. However, if multiple sensors are arranged in a suitable configuration and if specific properties of the disturbers, such as known polarization or time dependence are used, it is possible to remove disturbances generated by multiple sources.

Two magnetometers represent the minimal configuration needed to eliminate stray spacecraft magnetic fields. Many spacecraft carry two magnetometers attached at different positions along one boom. If the boom is long enough such that the distances between the disturbance sources are much smaller compared to the distances to the measurement points and if the disturbances have all either pure dipole or quadrupole character, then their $\mathcal{T}$ matrices will be the same and their collective disturbance can be separated from the ambient field in one step using only two sensors as it was done e.g. by Ness et al. (1974). Of course, any collection of dipoles will generate multipole moments. For the procedure to work, the quadrupole contribution must be much weaker than the dipole contribution at both sensors. If however, both dipole and quadrupole contributions are present at the same time with comparable strengths, then their $\mathcal{T}$ matrices will differ due to the different attenuation with the distance. In this case, one must rely on specific properties of the disturbance to eliminate the quadrupole contribution.

In contrast to the minimum two magnetometer configuration, one can imagine a configuration such as for each disturber there is a sensor placed much closer to it than to all other disturbers, plus an additional sensor far away from all disturbers. Then for each sensor the far disturbers can be assimilated to the ambient field and the problem becomes the single disturber problem discussed at the beginning of the section. Each contribution can then be separated from the ambient field independently. Such a sensor configuration is ideal and can be attained with a number of sensors placed within the spacecraft plus one sensor placed on a short boom.



If the disturbing magnetic field has a time dependent magnitude but does not change its direction, i.e. its variation is linearly polarized, then up to three independent, mutually orthogonal, simultaneously active disturbances can be separated using two sensors. This is done by projecting Eq. (7) on the direction of each disturbance. The direction of each disturbance can be deter-

mined using principal component analysis as described in sections 3 and 4. This kind of linear polarized disturbances produced by fixed configuration time-dependent currents are commonly encountered. The PiCoG cleaning method assumes this type of linearly polarized disturbances. If more than three disturbances, or non-linearly polarized disturbances, or disturbances with their polarization directions not mutually orthogonal are present, then information from more sensors is necessary. Different sensor pairs will correct different disturbances.

The SOSMAG configuration on board of the GK2A spacecraft lies somewhere in between the ideal configuration above and the minimum two magnetometers configuration. It consists of two high accuracy magnetometers placed on a relatively short boom and a number of resource saving magnetometers placed inside the spacecraft. As we will show in section 4, this configuration is well suited to apply the PiCoG cleaning method.

## 3    The Principal Component Gradiometer algorithm

The PiCoG cleaning algorithm is based on the fact that while the ambient magnetic field does not change over the spacecraft scale, the magnitude of a spacecraft generated disturbance in the magnetic field decreases with the distance to the disturbance source. Therefore, the disturbance can be recovered – and subsequently removed from the useful signal – by comparing measurements from sensors placed at different distances to the disturbance source as outlined in section 2. Here we describe the derivation of the transformation matrices $\mathcal{T}$ which allows the separation of disturbances generated by the spacecraft under

certain assumptions, but without prior knowledge about the positions of the disturbance sources.

The magnetic field measured by the sensor $i$ can be written as the sum of the ambient magnetic field, $\boldsymbol{B}(t)$, the disturbance $\sum_{q=1}^{N} \boldsymbol{b}^q(t, \boldsymbol{r}^{iq}) = \sum_{q=1}^{N} \boldsymbol{b}^{qi}(t)$ created by $N$ disturbance sources placed at relative positions $\boldsymbol{r}^{iq} = \boldsymbol{r}^i - \boldsymbol{r}^q$ from the sensor $i$ and a term containing the sensor specific disturbance (noise and time dependent offset), $\boldsymbol{Z}^i(t)$:

$$\boldsymbol{B}^{0,i}(t) = \boldsymbol{B}(t) + \sum_{q=1}^{N} \boldsymbol{b}^{qi}(t) + \boldsymbol{Z}^i(t) \tag{8}$$

where the index zero indicates the initially measured magnetic field.

We can eliminate the ambient field by subtracting the measurements from two sensors placed at different positions:

$$\Delta \boldsymbol{B}^{0,ij}(t) = \boldsymbol{B}^{0,i}(t) - \boldsymbol{B}^{0,j}(t) = \sum_{q=1}^{N} \Delta \boldsymbol{b}^{qij}(t) + \Delta \boldsymbol{Z}^{ij}(t) \tag{9}$$

To remove the disturbance in the measured data we have to find the correct coefficients $A_{kl}^{ij}$ for a linear combination of the components of the difference $\Delta \boldsymbol{B}^{0,ij}(t)$ between the measured magnetic field at each sensor position:

$$\boldsymbol{B}_{\text{corrected}}^i(t) = \boldsymbol{B}^{0,i}(t) + \mathcal{A}^{ij} \Delta \boldsymbol{B}^{0,ij}(t) \tag{10}$$





If we neglect the sensor specific disturbances, for single disturbers the matrix $-\mathcal{A}^{ij}$ is the same as the matrix $(\mathcal{T}(\boldsymbol{r}^i, \boldsymbol{r}^j) - \mathcal{I})^{-1}$ in Eq. (7). This may of course be computed if we know the exact coordinates of the sensors and of the disturbers and if the disturbers are pure single dipoles or quadrupoles. This is in general not true, therefore we will derive the correction matrix $\mathcal{A}^{ij}$ directly from the measurements.

## 3.1 First order correction

We now assume that one of the terms in Eq. (9) is much larger than the others. This is true if one of the disturbance sources is much stronger or much closer to one of the sensors than to the others. In this case, equations (8) and (9) are reduced to the single disturber form:

$$\boldsymbol{B}^{0,i}(t) = \boldsymbol{B}(t) + \boldsymbol{b}^i(t) + \boldsymbol{Z}^i(t) \tag{11}$$

$$\Delta\boldsymbol{B}^{0,ij}(t) = \Delta\boldsymbol{b}^{ij}(t) + \Delta\boldsymbol{Z}^{ij}(t) \tag{12}$$

where we drop the disturbance source index, $q$.

For many spacecraft, including GK2A, artificial disturbances keep their direction fixed at a given sensor position and only their magnitudes vary in time proportional to their (instantaneous) magnetic moment. Therefore, in the proper coordinate system, only one component of the measured field is affected by one disturbance source. However, since in general the direction of the disturbance varies from sensor to sensor, different reference systems must be used for different sensors.

To find the direction of the disturbance at the sensors positions we need to assume that the variance due to the disturbance at the sensor positions determines the maximum variance direction of the measured magnetic field. This holds either when the variance of the disturbance is much larger than the variance of the ambient field or when the variance of the ambient field does not have a preferred direction. In this case, the direction of the disturbance at both sensors can be estimated through variance analysis and the components of the magnetic field at the sensor $i$, corrected using measurements from the sensor $j$, can be written in the variance principal system (VPS) of the sensor $i$ measurements as:

$$B_x^{1,ij} = B_x^{0,i} - \alpha^{0,ij}(\Delta\boldsymbol{B}^{0,ij})_x \tag{13a}$$

$$B_y^{1,ij} = B_y^{0,i} \tag{13b}$$

$$B_z^{1,ij} = B_z^{0,i} \tag{13c}$$

The superscript "1" in equations (13) stands for the first order correction. Note that while the left hand sides and the first term of the right hand sides of equations (13) are represented in the VPS of the measurements at the sensor $i$, $(\Delta\boldsymbol{B}^{0,ij})_x$ in the right hand side of Eq. (13a) is represented in the VPS of the difference $\Delta\boldsymbol{B}^{0,ij}$. The VPS has the $x$-axis aligned with the maximum variance and the $z$-axis aligned with the minimum variance.

A first estimation of the $\alpha^{0,ij}$ factor is given by the variance of the measurements:

$$\alpha^{0,ij} = \pm\sqrt{\frac{\mathrm{Var}\left((\boldsymbol{B}^{0,i})_x\right)}{\mathrm{Var}\left((\Delta\boldsymbol{B}^{0,ij})_x\right)}} \tag{14}$$





The $\pm$ sign above is due to the fact that while the orientation of the $x$ axis of the VPS is determined from variance analysis, its sense remains arbitrary.

If $\mathcal{R}^{0,i}$ is the rotation matrix from the sensor system to the VPS of the measurements from the sensor $i$, and $\mathcal{R}^{0,ij}$ is the rotation matrix from the sensor system to the VPS of the difference $\Delta \boldsymbol{B}^{0,ij}$, then in the sensor system equations (13) take the form:

$$B_k^{1,ij} = B_k^{0,i} - \alpha^{0,ij}\left(\left(\mathcal{R}^{0,i}\right)^{-1}\right)_{kx}\left(\mathcal{R}^{0,ij}\Delta\boldsymbol{B}^{0,ij}\right)_x \qquad ; k = 1,\dots,3 \tag{15}$$

In matrix form the above relation can be written as:

$$\boldsymbol{B}^{1,ij} = \boldsymbol{B}^{0,i} + \mathcal{A}^{0,ij}\Delta\boldsymbol{B}^{0,ij} \tag{16}$$

where the matrix $\mathcal{A}$ with elements

$$A_{kl}^{0,ij} = -\alpha^{0,ij}\left(\left(\mathcal{R}^{0,i}\right)^{-1}\right)_{kx}\left(\mathcal{R}^{0,ij}\right)_{xl} \tag{17}$$

is the correction matrix for the first (strongest) disturber. Note that there is no implicit summation over repeating indices.

### 3.1.1 Collinear case

While not required, the special case when the disturbance source is collinear with the two sensors is instructive. In this case, the direction of a linearly polarized disturbance will be the same at both sensors, therefore the same coordinate system will be used for equations (13). Substituting $B_x^{0,i}$ in the Eq. (13a) using Eq. (11), we obtain:

$$B_x^{1,ij} = B_x + \left(a - \alpha^{0,ij}(a-1)\right)b_x^j + Z_x^i - \alpha^{0,ij}\left(Z_x^i - Z_x^j\right) \tag{18a}$$

$$B_x^{1,ji} = B_x + \left(1 + \alpha^{0,ji}(a-1)\right)b_x^j + Z_x^j - \alpha^{0,ji}\left(Z_x^j - Z_x^i\right) \tag{18b}$$

where we made use of the proportionality between the spacecraft generated disturbances at the sensors $i$ and $j$: $b_x^i = ab_x^j$. For a dipolar disturber at distance $r^i$ from the sensor $i$, and $r^j$ from the sensor $j$, $a = (r^j/r^i)^3$. For a quadrupolar disturber $a = (r^j/r^i)^4$.

To eliminate the disturbance $b_x^j$, the factor in front of it must vanish, therefore

$$\alpha^{0,ij} = \frac{a}{a-1} \qquad \text{and} \qquad \alpha^{0,ji} = \frac{-1}{a-1} \tag{19}$$

This shows that in the collinear case, the sum of the $\alpha$ coefficients is equal to one:

$$\alpha^{0,ij} + \alpha^{0,ji} = 1 \tag{20}$$

A consequence of the above is that the difference between the corrected measurements at the two sensors is always zero:

$$\Delta B_x^{1,ij} = B_x^{1,ij} - B_x^{1,ji} = \left(1 - \alpha^{0,ij} - \alpha^{0,ji}\right)\left(b_x^i - b_x^j + Z_x^i - Z_x^j\right) \equiv 0 \tag{21}$$





In other words, the corrected field is the same regardless which sensor is used as "primary" sensor: $B_x^{1,ij} = B_x^{1,ji}$

For $\alpha$ obeying Eq. (19) the corrected field given by equations (18) is:

$$B_x^{1,ij} = B_x + Z_x^i - \alpha^{0,ij}\left(Z_x^i - Z_x^j\right) \tag{22}$$

Comparing the above with Eq. (11) shows that, apart from eliminating the spacecraft generated disturbance $b_x^i$, the procedure introduces an additional disturbance which mixes the two sensor specific disturbances $Z_x^i$ and $Z_x^j$ scaled by $\alpha^{0,ij}$, potentially increasing the noise in the corrected measurements. This effect was also noted by Delva et al. (2002). However, if $\alpha^{0,ij}$ approaches unity (disturbance source much closer to sensor $i$), the $i$ sensor specific noise is replaced by the $j$ sensor specific noise which might lead to reduced noise.

**3.2    Higher order corrections**

Further corrections can be iteratively applied as long as the maximum variance directions of the disturbances do not coincide. The iteration relation from order $n-1$ to order $n$ is

$$\boldsymbol{B}^{n,ij} = \boldsymbol{B}^{n-1,ij} + \mathcal{A}^{n-1,ij}\Delta\boldsymbol{B}^{n-1,ij} \quad ; \quad \boldsymbol{B}^{0,ij} = \boldsymbol{B}^{0,i} \tag{23}$$

with

$$A_{kl}^{n-1,ij} = -\alpha^{n-1,ij}\left(\left(\mathcal{R}^{n-1,i}\right)^{-1}\right)_{kx}\left(\mathcal{R}^{n-1,ij}\right)_{xl} \tag{24}$$

The $\alpha^{n,ij}$ coefficient is estimated from the variance of the field corrected up to order $n$. The rotation matrices $\mathcal{R}^{n,i}$ and $\mathcal{R}^{n,ij}$ refer to the order $n$ corrected field.

Using Eq. (23) and Eq. (24) we find the corrected magnetic field in the second and third order written as linear combinations of the difference of the measurements taken at the two sensors:

$$\boldsymbol{B}^{2,ij} = \boldsymbol{B}^{0,i} + \left(\mathcal{A}^{0,ij} + \mathcal{A}^{1,ij} + \mathcal{A}^{1,ij}\left(\mathcal{A}^{0,ij} + \mathcal{A}^{0,ji}\right)\right)\Delta\boldsymbol{B}^{0,ij} \tag{25}$$

$$\begin{aligned}\boldsymbol{B}^{3,ij} = \boldsymbol{B}^{0,i} + \Big(&\mathcal{A}^{0,ij} + \mathcal{A}^{1,ij} + \mathcal{A}^{2,ij}\\ &+ \mathcal{A}^{1,ij}\left(\mathcal{A}^{0,ij} + \mathcal{A}^{0,ji}\right) + \mathcal{A}^{2,ij}\left(\mathcal{A}^{0,ij} + \mathcal{A}^{0,ji} + \mathcal{A}^{1,ij} + \mathcal{A}^{1,ji}\right)\\ &+ \mathcal{A}^{2,ij}\left(\mathcal{A}^{1,ij} + \mathcal{A}^{1,ji}\right)\left(\mathcal{A}^{0,ij} + \mathcal{A}^{0,ji}\right)\Big)\Delta\boldsymbol{B}^{0,ij}\end{aligned} \tag{26}$$

The corrected field $\boldsymbol{B}^{n,ij}$ determined for the sensor $i$ can replace now the measured field $\boldsymbol{B}^{0,i}$ in a similar procedure involving the next (third) sensor, until the measurements from all sensors are used.

Ideally, the hardware should consist of a "main", least disturbed sensor and additional sensors close to each major disturbance source as described in section 2.2. Then, only the first order correction for each sensor pair containing the main sensor is necessary to clean the data. However, also other sensor configurations can be used as described in the next section.

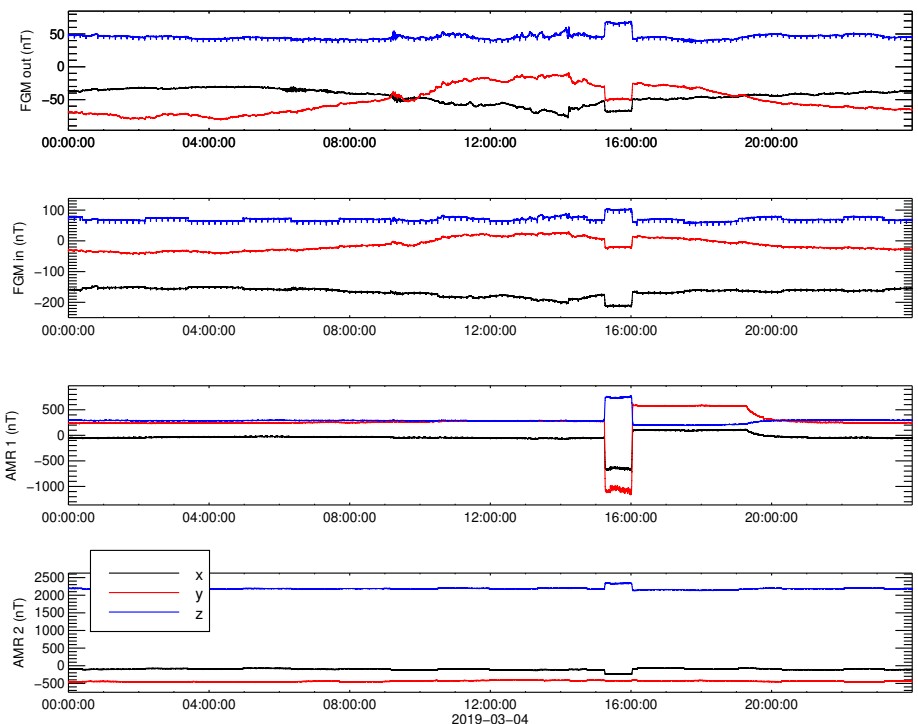

**Figure 1.** The components in the OB sensor system of the uncorrected measurements taken by the four magnetometers onboard GK2A on March 4 2019. From top to bottom: FGMO, FGMI, AMR1, and AMR2.

## 4 Application to GK2A SOSMAG measurements

The GK2A spacecraft launched on December 4 2018 on a 128.2° East geostationary orbit is operated by the Korea Aerospace Research Institute (KARI) and provides meteorological and space weather monitoring over the Asia-Pacific region. The magnetic field vector is measured by the SOSMAG instrument (Magnes et al., 2020) at four locations onboard the spacecraft: two high accuracy 3-axis Flux Gate Magnetometers (FGM ) (Primdahl, 1979; Acuña, 2002) placed at the end (outboard sensor, FGMO) and respectively 80 cm from the end (inboard sensor, FGMI) of an approximatively one meter long boom, and two 3-axis Anisotropic Magnetic Resistance (AMR) (Brown et al., 2012) solid state magnetometers placed within the body of the spacecraft.

The placement of the sensors can be seen in Fig. 6 of Magnes et al. (2020). Compared with the spacecraft dimensions, $(290 \times 240 \times 460)$ cm, the magnetometer boom is relatively short, leading to strong spacecraft-generated disturbances at both FGM sensors.

As far as magnetic cleanliness is concerned, GK2A is a black box, i.e. no access to spacecraft operation time tables and satellite specific housekeeping data is available to aid the cleaning of the magnetic field data. Therefore the cleaning process must be based exclusively on the magnetic field measurements. Our goal is to eliminate the time dependent spacecraft generated





disturbances from the FGM measurements. The strategy we adopt in order to take maximum advantage of the high accuracy of the FGMs and of the placement of the AMRs close to the disturbance sources, is to first use the AMR measurements to clean the data from both FGMs, and then use these corrected measurements to clean each other.

When a disturbance is much stronger at one sensor – as it is the case for the AMR sensors – the scaling factor $\alpha$ is roughly given by the ratio of the magnitudes of the disturbance at the two sensors. This ratio is about 40 for AMR1, and 5 for AMR2, when paired with any of the FGMs. Since the sensor specific noise for the flux gate magnetometers is lower by a factor of 20 compared to the AMR sensor noise, according to Eq. (22), the correction using the AMR sensors will introduce roughly the AMR noise divided by $\alpha$. In particular, for the AMR2 one fifth of its noise would be introduced in the corrected measurements.

Since the same main disturbance is seen by both AMR sensors, no extra information is present in the AMR2 measurements, therefore we decided not to use the AMR2 sensor for removing the stray time dependent spacecraft magnetic field. The $1/40$ from the AMR1 noise is much more favourable therefore we will use this sensor to clean both FGM sensors measurements.

### 4.1   FGM outboard and FGM inboard cleaning using the AMR1

Figure 1 shows the uncorrected measurements taken by the outboard FGM, inboard FGM and the two AMR sensors on March
4 2019. We choose this day because it is representative for the routine operations, all the disturbance sources are active and the ambient field shows little variance. Both step-like and spike-like disturbances can easily be seen in the picture. Among them, a prominent step-like disturbance between about 15:00 and 16:00 is clearly detected by all four sensors, showing a very large magnitude at the AMR1. Note that the disturbance, which starts shortly after 15:00 affects the measurements until around 20:00. Because at 15:00 UT the spacecraft is close to local midnight we call this disturbance "midnight disturbance" (MD) to
distinguish it from the other step-like disturbances. In 2019 this disturbance appears daily at the beginning and at the end of the year for about 14 weeks in total. We begin the cleaning of the data by first removing this disturbance from the FGM sensors measurements using the AMR1 data.

For the sake of clarity, in the following we use the index $s$ for the outboard FGM, the index $t$ for the inboard FGM, and the index $a$ for the AMR1 sensor. Equation (16) giving the magnetic field measured by the FGM sensors corrected in the first order
using the AMR1 sensor yields:

$$\boldsymbol{B}^{1,sa} = \boldsymbol{B}^{0,s} + \mathcal{A}^{0,sa}\left(\boldsymbol{B}^{0,s} - \boldsymbol{B}^{0,a}\right) \tag{27a}$$

$$\boldsymbol{B}^{1,ta} = \boldsymbol{B}^{0,t} + \mathcal{A}^{0,ta}\left(\boldsymbol{B}^{0,t} - \boldsymbol{B}^{0,a}\right) \tag{27b}$$

with the matrices $\mathcal{A}^{0,ja}$; $j = s,t$ given by Eq. (17).

We select the time interval [15:10,16:15] to isolate the targeted disturbance and use it to calculate the correction matrices.
To lift the indetermination of the sign of the scaling factor $\alpha$ in Eq. (14) we compute the corrected fields Eq. (27) for both signs and keep the sign for which the disturbance is successfully removed. Equation (14) gives a very good estimation for the scaling factor. However, since this estimation uses the measured magnetic field which includes the ambient magnetic field, it may slightly deviate from the correct value. To improve the precision one may use the scaling factor determined from Eq. (14) as initial value for a minimization procedure of the correlation between $(\Delta\boldsymbol{B}^{0,ja})_x$ and the corrected $(\boldsymbol{B}^{1,ja})_x$. While we





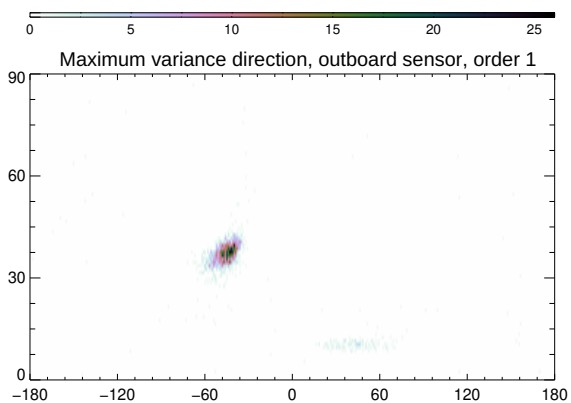

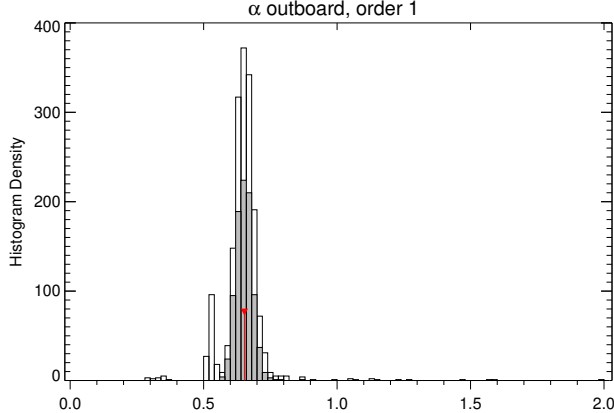

**Figure 2.** The cleaning parameters resulting from the sliding window scan for the first order correction of the outboard FGM. The left panel shows the number density of the maximum variance direction $(\theta_w, \varphi_w)$ on a $1° \times 1°$ grid. The right panel shows a histogram of the statistical distribution of the $\alpha_w$ coefficients. The grey filled bars are the coefficients corresponding to directions within $2.5°$ from most probable direction. The red vertical line marks their mean value.

found this to improve the determination of $\alpha$ for FGMI-FGMO cleaning for days with disturbed ambient magnetic filed, for the AMR1 cleaning on March 4 2019, the minimization does not change significantly the value of $\alpha$.

The angle between the direction of the disturbance at the AMR1 sensor and the direction of the disturbance at the inboard FGM sensor is $31°$. For the outboard FGM sensor this angle is $25°$ indicating that the disturbance source is not collinear with either of the sensor pairs. This is not surprising given the placement on the spacecraft body of the AMR sensors. Even so, the

sum of the $\alpha$ coefficients differs from unity with less than 0.005.

The higher order corrections should identify and eliminate disturbances roughly ordered by their strength at the AMR1 location. However, attempting the second order correction only introduces spurious data in the FGMs measurements, increasing their variance. This is because the noise level of the AMR sensors is higher than the noise level of the FGM sensors and the AMR1 noise is added to the corrected measurements according to Eq. (22). Consequently we limit the AMR1 corrections to

the first order.

Since the data cleaning onboard the spacecraft should not require frequent updates of the correction parameters once uploaded to the spacecraft, it is necessary that the determined $\mathcal{A}$ correction matrices remain stable in time. In order to confirm this we checked the stability of the cleaning parameters by using the same procedure once for every week showing the targeted disturbance in 2019. The standard deviation for the maximum variance directions was below $1°$, while the standard deviation

for the scale factors was below $10^{-3}$. These low values are not surprising since for a given source the cleaning parameters depend only on its multipole character and on the geometry of the sources-sensors system. Other factors such as the intensity of the current generating the magnetic disturbance or the temperature do not influence the cleaning parameters. Applying the correction using the determined set of parameters removes the disturbance throughout the entire 2019 year.



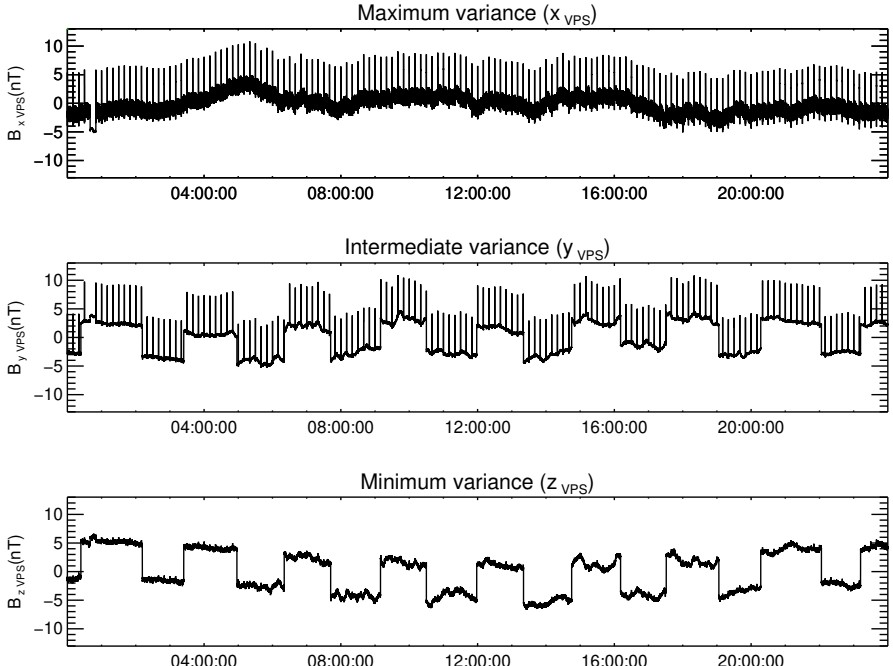

**Figure 3.** The difference $\Delta B^{st}$ represented on components in its own VPS before the first order correction was applied. The mean values were subtracted from all components.

## 4.2 FGM cleaning using the AMR1-corrected data

We now use the AMR1-corrected FGMO and FGMI measurements Eq. (27) as starting point in the iteration Eq. (23) for cleaning the FGMO data using the FGMI data and vice-versa.

Unlike the single step disturbance we dealt with in section 4.1, the disturbances to be removed now show a repetitive pattern over the entire day, apparent in Fig. 1. Apart from the removed large magnitude disturbance, one can visually identify at least two other types of disturbances in Fig. 1: step-like disturbances at a time scale of over one hour, and spike-like disturbances at 300 time scales of minutes. To determine the correct cleaning parameters, the length of the analysis interval has to be chosen such as to contain many samples of the targeted disturbance but avoid including other disturbances. This can be accomplished by eliminating first the highest frequency disturbances using small enough interval length. Then the interval length is increased to encompass the next frequent disturbance.

In order to increase the precision of the cleaning and to have an indication on the stability of the determined parameters 305 we compute the cleaning parameters using sliding windows covering the entire 24 h interval. For each window $w$ we find the scaling factor $\alpha_w$, the elevation angle $\theta_w$ and azimuth angle $\varphi_w$ of the maximum variance direction. After we scan the entire day interval, we determine the most probable direction $(\theta, \varphi)$ of the maximum variance which determines the rotation matrices

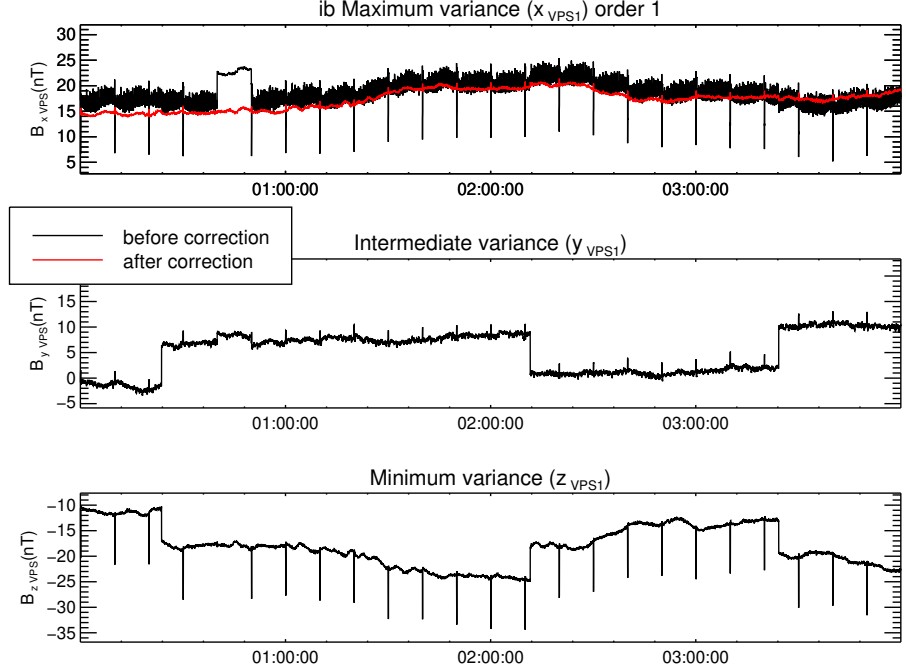

**Figure 4.** The initial AMR1-corrected FGM inboard measurements represented in the inboard VPS are plotted with the black lines. The first order correction is plotted with red. Mean values were subtracted.

$\mathcal{R}$ in Eq. (24). For this direction we select the corresponding coefficients $\alpha_w$ and we compute their average value. At the end, the correction matrix $\mathcal{A}$ is computed using $\theta, \varphi$ and $\alpha$.

The disturbances can be much better identified in the difference $\Delta \boldsymbol{B}^{0,sta} = \boldsymbol{B}^{0,sa} - \boldsymbol{B}^{0,ta}$ plotted in Fig. 3. The difference was first rotated in the VPS corresponding to a window length of $100\,\mathrm{s}$, smaller than the time interval between the spike-like disturbances. In this coordinate system, different disturbance types tend to sort themselves on components.

The spike-like disturbances appear now in the $x$ and $y$ components with a cadence of $10\,\mathrm{min}$ and an magnitude larger than $10\,\mathrm{nT}$ during the entire interval. The step-like disturbances with slightly smaller magnitudes than the spikes are present in the

$y$ and $z$ components. The duration between upward and downward variations of the step-like disturbances is $80\,\mathrm{min}$ to $90\,\mathrm{min}$, not as regular as the timing for the spikes. A new type of disturbance, not evident in Fig. 1 is now clearly apparent as a variation at higher frequencies (periods less than one minute) than the steps or the spikes cadence. A closer investigation shows that this disturbance is irregular, with a maximum peak to peak amplitude of up to $4\,\mathrm{nT}$ in the $x$ component and with its spectral power spread up to the Nyquist frequency.

The much smaller amplitude of the higher frequency disturbance in the $y$ and $z$ components indicate its linear polarization. This was the disturbance which determined the orientation of the VPS used to plot the differences in Fig. 3. However, the spike-like disturbance has a large contribution on the $x$-component, therefore it is not orthogonal to the high-frequency disturbance.





In fact, the angle between the maximum variance directions of the spike-like disturbance and of the high-frequency disturbance is 25°, which grossly violates the orthogonality condition. As a consequence, if the two sources producing the high-frequency and the spike-like disturbances have different scaling factors, the PiCoG method will not be able to remove both disturbances from the $x$-component using one single pair of sensors. The 75° angle between the directions of the spike-like disturbance and the step-like disturbance is more favourable but it will still prevent the complete removal of these disturbances simultaneously unless they have the same scaling factors. The closest to orthogonality is the angle between the directions of the high-frequency disturbance and of the step-like disturbance, which is 87°. Since the orthogonality condition is not fulfilled, to proceed further we must assume that the disturbances to be removed come from a small volume compared with the distances between the sensors and therefore their scaling factors are not very different from each other. The results of the cleaning will either confirm or infirm our assumption.

For the first order correction we target the highest frequency disturbance by choosing the same window length of 100 s used to compute the VPS for the difference plotted in Fig. 3. The statistical distribution for the resulted direction $(\theta_w, \varphi_w)$ of the maximum variance and a histogram of the $\alpha_w$ values is shown in Fig. 2. Both distributions exhibit clear isolated maxima which is a strong indication that the targeted disturbance does not change its characteristics during the day interval. The angle between the disturbance directions at the two sensors is 15°, closer to collinearity than for the AMR1 correction.

Since the disturbances are larger at the inboard sensor, the effect of the correction is better illustrated for it than for the outboard sensor. The first order correction of the inboard measurements for the first four hours of the day is plotted in Fig. 4 with red over the initial AMR-corrected inboard measurements represented in the inboard VPS. The targeted high frequency disturbance is eliminated from the $x$ component. As apparent from the top panel of Fig. 4, between 00:40 and 00:50 the high frequency disturbance was switched off. One can see that the disturber also introduces a constant offset of about 5 nT which is removed by the applied correction.

The magnitude of the spike-like disturbance is much reduced in the $x$ component of the corrected magnetic field in Fig. 4 so we conclude that the sources of both high frequency and step-like disturbances are close to each other and are therefore removed together from the maximum variance component. This justifies the application of the PiCoG method in this particular case when the directions of the two disturbances are far from orthogonal.

For the second order correction we target the remaining spike-like disturbance by choosing a window width of 700 s. Figure 5 shows the result of the second order correction for the inboard sensor. Both the targeted spike-like disturbance and the step-like disturbance are removed from the $x$ component by this correction step showing that indeed the distances between the sources of all three disturbances are much smaller than the distances between the disturbance sources and the FGM sensors, confirming our previous assumption.

The step-like disturbance and traces of the spike-like disturbance still remain in the $y$ and $z$ components in Fig. 5. To eliminate them we select a window width of 16 000 s, enough to always include at least one step-like disturbance sample. As seen in Fig. 6, the correction removes the targeted disturbance and strongly reduces the remnants of the other two disturbance types from the $x$ component. A leftover step-like disturbance, with an magnitude of about 1 nT is still visible in the intermediate variance component. This is due to the fact that, even with carefully chosen window lengths, the maximum variance directions

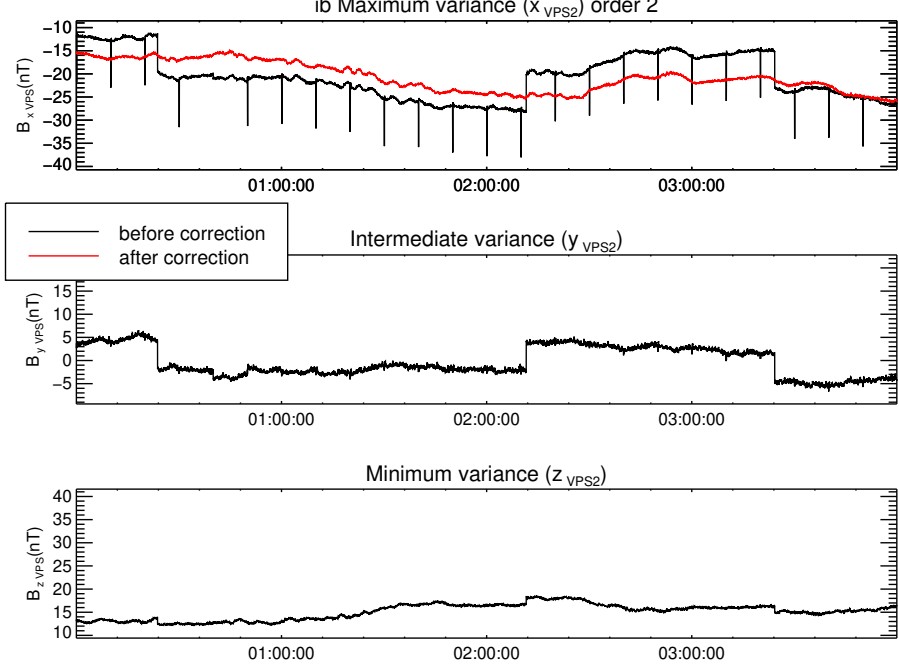

**Figure 5.** The first (black) and the second order correction (red) for the inboard FGM sensor. Mean values were subtracted.

are still influenced by all present disturbances, therefore do not perfectly coincide with the polarization direction of the targeted disturbances. This leads to remanent disturbance on the other components. In our case, leftovers from the high frequency dis-

turbance interfered with the determination of the step-like disturbance polarization direction. The result is the further reduction of the high frequency disturbance at the cost of not completely removing the step-like disturbance.

To check the stability of the cleaning parameters we determine them for every Sunday in 2019 with available data. The procedure produces very similar results apart from three instances when the ambient magnetic field was very disturbed. After eliminating the three outliers we computed the standard deviations for the principal component directions and for the scale

factors, displayed in Table 1. The table also shows the corresponding maximum change in the corrected magnetic field on 2019.03.04 due to changes in the parameters equal to the standard deviations. The last row displays the maximum change due to the deviations in the parameters for one single order while the parameters for the other orders are kept constant. Similarly, the last column displays the maximum change related to variations either in one single direction or in one scale factor. The last value in the table is the maximum change in the corrected magnetic field corresponding to all computed deviations,

$B_{\max}^{\mathrm{dev}} = 0.186\,\mathrm{nT}$. This is the expected error due to the variations in the ambient magnetic field. However, the main error source is related with disturbers which do not fit our assumptions such as collinearity or with the presence of higher multipoles, as discussed in section 5.





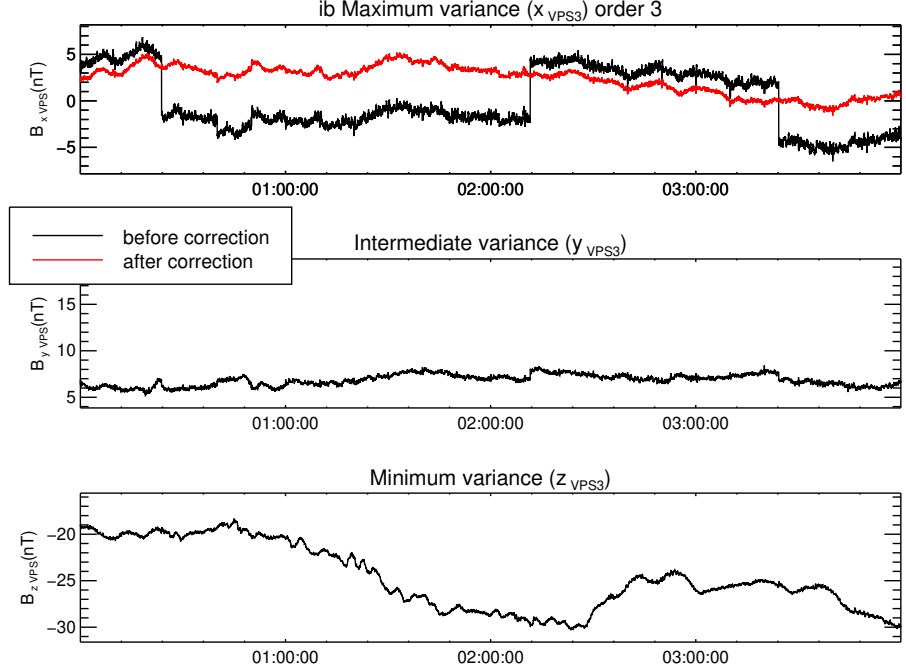

**Figure 6.** The second (black) and the third order correction (red) for the inboard FGM sensor. Mean values were subtracted.

| | order 1 | order 2 | order 3 | $\max(B_{\text{dev}})$ (nT) |
|---|---|---|---|---|
| direction OB (deg) | 0.056 | 0.641 | 3.987 | 0.068 |
| direction IB (deg) | 0.066 | 0.265 | 0.625 | 0.105 |
| direction $\Delta$B (deg) | 0.080 | 0.180 | 0.206 | 0.035 |
| scaling factor OB | 0.006 | 0.003 | 0.013 | 0.093 |
| scaling factor IB | 0.005 | 0.003 | 0.013 | 0.077 |
| $\max(B_{\text{dev}})$ (nT) | 0.064 | 0.082 | 0.114 | 0.186 |

**Table 1.** Standard deviations for FGMI-FGMO correction directions and scale factors, together with the corresponding maximum deviation of the corrected magnetic field.

The standard deviations for the first two orders are very small, indicating very stable cleaning parameters for the high frequency disturbance and for the spike disturbance. The third order, used to clean the step-like disturbance shows larger

deviations, especially for the outboard maximum variance direction. This is because of the low contribution of the step-like disturbance at the outboard sensor which makes the procedure susceptible to the influence of the ambient magnetic field.





### 4.3 Parameters for spacecraft upload

Since the onboard correction is designed as a one-step linear combination of the measurements from different sensors, it cannot follow the iterative procedure described in section 4. Therefore, we have to write the final correction in the form:

$$\boldsymbol{B}^{c,s} = \mathcal{M}^s \boldsymbol{B}^{0,s} + \mathcal{M}^t \boldsymbol{B}^{0,t} + \mathcal{M}^a \boldsymbol{B}^{0,a} + \boldsymbol{G}^s \tag{28}$$

where the superscript $c$ stands for the combined correction, and the matrices $\mathcal{M}^j$ have constant coefficients given by the $\mathcal{A}$ correction matrices determined on ground using the iterative procedure:

$$\mathcal{M}^s = \left(\mathcal{I} + \mathcal{C}^s\right)\left(\mathcal{I} + \mathcal{A}^{0,sa}\right) \tag{29a}$$

$$\mathcal{M}^t = -\mathcal{C}^s\left(\mathcal{I} + \mathcal{A}^{0,ta}\right) \tag{29b}$$

$$\mathcal{M}^a = \mathcal{C}^s \mathcal{A}^{0,ta} - \left(\mathcal{I} + \mathcal{C}^s\right)\mathcal{A}^{0,sa} \tag{29c}$$

Here $\mathcal{I}$ denotes the identity matrix and the matrix $\mathcal{C}^s$ has the form:

$$
\begin{aligned}
\mathcal{C}^s = {} & \mathcal{A}^{0,st} + \mathcal{A}^{1,st} + \mathcal{A}^{2,st} \\
& + \mathcal{A}^{1,st}\left(\mathcal{A}^{0,st} + \mathcal{A}^{0,ts}\right) + \mathcal{A}^{2,st}\left(\mathcal{A}^{0,st} + \mathcal{A}^{0,ts} + \mathcal{A}^{1,st} + \mathcal{A}^{1,ts}\right) \\
& + \mathcal{A}^{2,st}\left(\mathcal{A}^{1,st} + \mathcal{A}^{1,ts}\right)\left(\mathcal{A}^{0,st} + \mathcal{A}^{0,ts}\right)
\end{aligned}
\tag{30}
$$

The AC correction described in section 4.2 introduces a constant offset in the corrected data. This corresponds to the sources whose disturbances were removed. In practice however, there are additional DC offsets affecting the measurements which are
treated in a separate cleaning step. The vector $\boldsymbol{G}^s$ can be used to restore the original DC offset if a pure AC correction is desired.

$$\boldsymbol{G}^s = \langle \boldsymbol{B}^{0,s} \rangle - \left(\mathcal{M}^s \langle \boldsymbol{B}^{0,s} \rangle + \mathcal{M}^t \langle \boldsymbol{B}^{0,t} \rangle + \mathcal{M}^a \langle \boldsymbol{B}^{0,a} \rangle\right) \tag{31}$$

From equations (29) results that the sum of the $\mathcal{M}$ matrices is equal to the unit matrix:

$$\mathcal{M}^s + \mathcal{M}^t + \mathcal{M}^a = \mathcal{I} \tag{32}$$

A consequence of Eq. (32) is that an arbitrary vector added to the measurements $\boldsymbol{B}^{0,s}, \boldsymbol{B}^{0,t}, \boldsymbol{B}^{0,a}$ in the expression of the offset $\boldsymbol{G}^s$ in Eq. (31) vanishes, therefore $\boldsymbol{G}^s$ is independent on the ambient magnetic field. This is to be expected because the magnetic field measurements enter the correction only as differences between distinct sensors hence the correction – therefore also the offset due to the correction – is determined only by the spacecraft generated disturbances. This makes $\boldsymbol{G}^s$ a useful tool for monitoring changes in the DC offsets.
Applying Eq. (28) to the FGMO measurements yields the combined AMR1 - FGMI correction to the outboard FGM measurements. We plot the original outboard FGM measurements in sensor system with black lines and the result of the combined correction with red lines in Fig. 7.

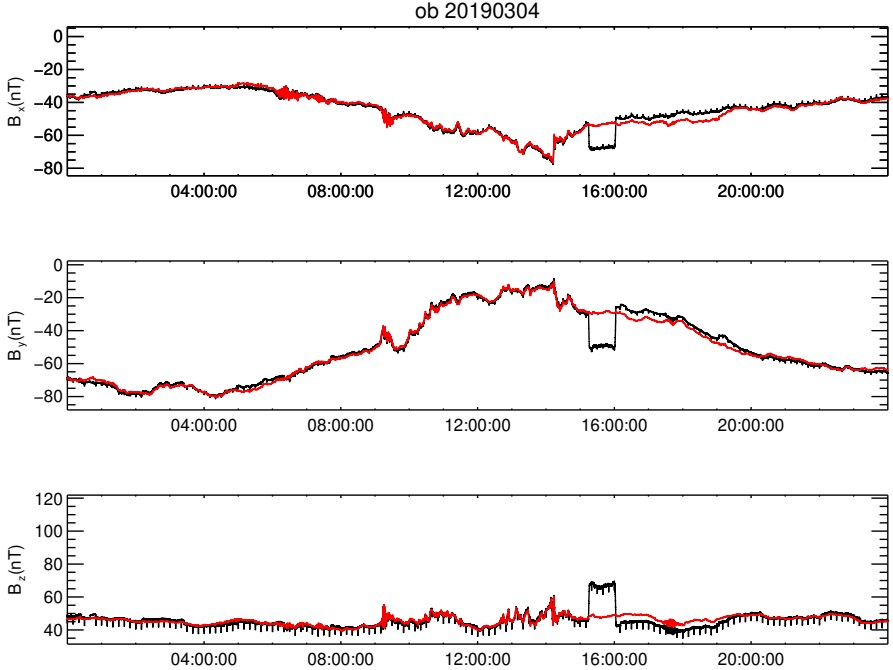

**Figure 7.** The final combined correction result for 2019.03.04 in the sensor system. The black lines show the original measurements taken by the outboard FGM, the red lines show the corrected data. The DC offset was restored to the value before the correction.

The $\mathcal{M}$-matrices were uploaded on GK2A four months after its launch. Since then the magnetic field measurements are corrected onboard and transmitted to the ground stations within minutes from acquisition. The stability of the correction
parameters is monitored and a new set of parameters will be computed and uploaded in case changes in the spacecraft operation will require a change in the parameters.

## 5   Errors and limitations

Even though we were able to eliminate most of the magnetic field disturbances onboard the GK2A spacecraft, we need to be aware of the limitations the proposed method is subject to. We have already seen that due to other disturbances or to the ambient
magnetic field variations, the maximum variance direction might not coincide with the polarization direction of the disturbance to be removed. This slight difference will cause non-zero projections of the disturbance on the intermediate and minimum variance direction components which are not removed by the current applied correction. They will be reduced however by the next correction if the targeted sources lie close to each other. A disturbed ambient magnetic field may also interfere with the determination of the scaling factors. While there are ways to mitigate these effects, they are not within the scope of the present
work.



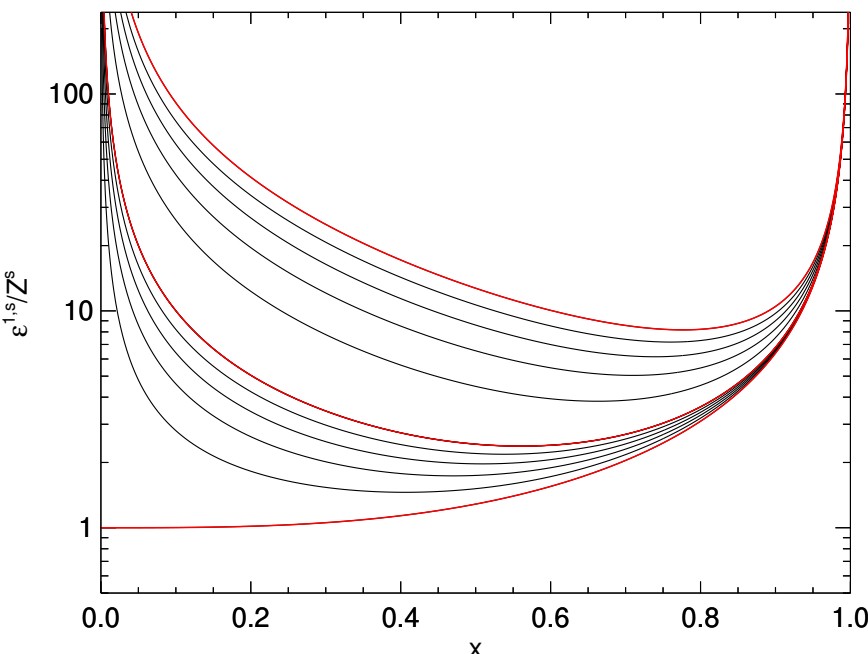

**Figure 8.** The error due to sensor specific disturbances and to the quadrupole disturbance introduced by PiCoG dipole disturbance correction. The $x$-axis represents the position $x = r^t/r^s$ of the inboard sensor relative to the outboard sensor. Sensor noise is the same for both sensors. Each line corresponds to a fixed value of the quadrupole disturbance at the outboard position. Red lines, bottom to top: $b_q^s = (0, 1 \text{ and } 10) \times Z^s$. The black lines in between correspond to $(0.2, 0.4, 0.6 \text{ and } 0.8) \times Z^s$ and $(2, 4, 6 \text{ and } 8) \times Z^s$, respectively.

An important benefit of the PiCoG method is the ability to treat up to three separate disturbance sources using measurements from two sensors. In order to be able to decouple the individual disturber contributions, two conditions must be satisfied: the disturbances must have well defined polarization directions and these directions must be orthogonal to each other. This may seem a strong condition to impose. However, apart from moving mechanisms such as reaction wheels, many, if not most of the

magnetic disturbances from a spacecraft come from current loops without phase delays and are therefore linearly polarized. The orthogonality on the other hand, is not guaranteed. Even in the non-orthogonal case, disturbances coming from sources close to each other compared to the distance to the sensors share the same scaling factor (if both are either dipoles or quadrupoles) and are therefore removed together. A possible way to treat non-orthogonal disturbances coming from positions separated by large distances compared to the distances to the sensors is first transforming the data to a non-orthogonal system with its axes

aligned with the maximum variance directions of the three largest disturbers. This exercise is left for future examination.

Another class of error sources are additional disturbances which do not follow the determined scaling factor $\alpha$ or that are present at one sensor only. Among these are the sensor specific noise, temperature effects which sometimes cause sensor





offset oscillations, and multipoles of higher order than the targeted disturbance. These disturbances are introduced into the cleaned magnetic field data either reduced or enhanced, depending on the sensor positions. In particular for GK2A, sensor

offset oscillations triggered by large temperature gradients are quite significant reaching peak to peak amplitudes up to 5 nT in the cleaned data (Magnes et al., 2020).

To estimate the error introduced by the sensor specific noise combined with a quadrupole contribution additional to a dipole disturbance to be removed, let us assume a simple collinear geometry: A disturber placed in the origin of the coordinate system producing a disturbance characterized by both a dipole moment $M$ and a quadrupole moment $\mathcal{Q}$, an inboard sensor placed

at the distance $r^t$ characterized by a sensor specific noise $Z^t$, and an outboard sensor placed at the distance $r^s$ characterized by a sensor specific noise $Z^s$. In these conditions, the correction of the dipole disturbance will introduce an error stemming from the quadrupole disturbance and the sensor specific disturbances. The magnitude of the error will depend on the relative positions of the two sensors, on the sensor specific noise and on the strength of the quadrupole disturbance. After projecting on the principal component direction, the magnetic field measured by the outboard sensor is (dropping the $x$ component index):

$$B^{0,s} = B + b_d^s + b_q^s + Z^s \tag{33}$$

where $b_d^s$ and $b_q^s$ represent the disturbance dipole and quadrupole contributions at the outboard sensor. A similar expression can be written for the inboard sensor.

The corrected field is obtained by applying equation (13a):

$$B^{1,s} = B + \epsilon^{1,s} \tag{34}$$

with the error $\epsilon^{1,s}$ given by

$$\epsilon^{1,s} = (1 - \alpha)Z^s + \alpha Z^t + (1 - \alpha)b_q^s + \alpha b_q^t \tag{35}$$

Making the notation

$$x = \frac{r^t}{r^s} < 1 \tag{36}$$

and keeping in mind that $b_d^s = x^3 b_d^t$ and $b_q^s = x^4 b_q^t$, equation (35) becomes:

$$\epsilon^{1,s} = \frac{1}{1 - x^3}\left[1 + x^3 \frac{Z^t}{Z^s} + \left(\frac{1}{x} - 1\right)\frac{b_q^s}{Z^s}\right]Z^s \tag{37}$$

Similar with Neubauer (1975) findings, the optimum position $x$ results from a trade off between the error due to the sensors, $Z$, and the error due to higher order multipoles, $b_q$. We plot the error given by equation (37) for a number of quadrupole strengths in Fig. 8. The bottom red line corresponds to zero quadrupole moment. In this case, minimum error, equal to the outboard sensor specific noise, is obtained for $x = 0$, i.e. for the inboard sensor placed at the position of the dipole disturber.

As soon as a a higher multipole is present, the inboard sensor must be moved away from the disturbance source in order to minimize the error. Already for a quadrupole disturbance at the outboard position equal to a tenths of the sensor noise, the

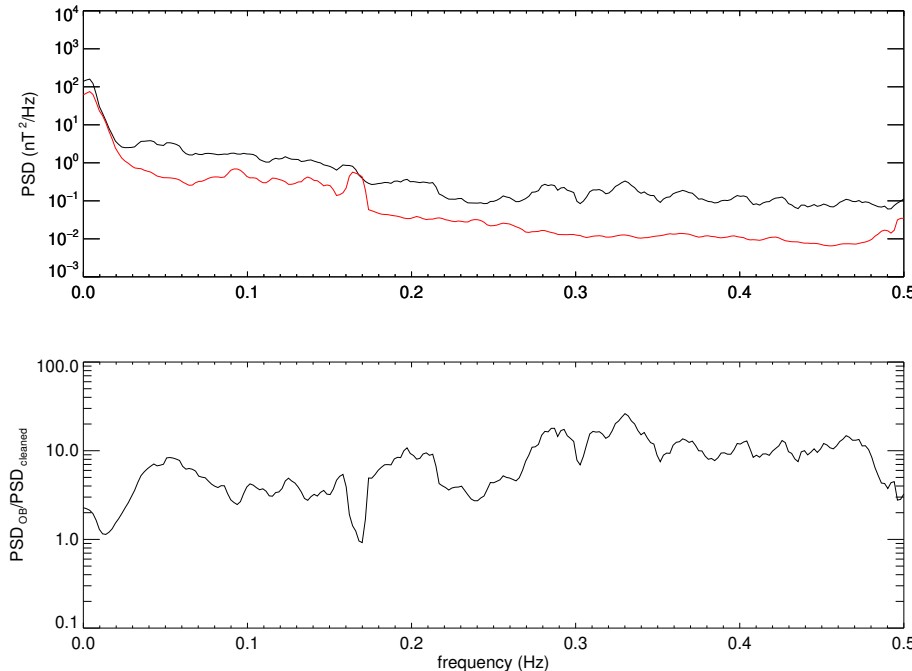

**Figure 9.** Top panel: Power spectral density of the initial FGMO measurements (black line) compared with the PSD of the cleaned data(red line). Bottom panel: the ratio between the initial and cleaned PSDs for 2019.03.04.

optimum position of the inboard sensor is almost at the mid distance between the disturber and the outboard sensor. When the quadrupole disturbance becomes equal with the sensor noise, the optimum distance becomes about $0.6x$ (mid red line). If the boom is very short, the quadrupole disturbance at the outboard sensor can reach very large values. The topmost red line in Fig. 8 corresponds to a quadrupole contribution ten times as large as the outboard sensor noise. In this case the optimum position of the inboard sensor approaches even more the outboard sensor position ($0.8x$).

A way to estimate the overall performance of the cleaning is to compare the power spectral densities of the initial measurements with the PSDs of the cleaned data as shown in Fig. 9. The spectra in the figure were computed as the average over the entire 2019.03.04 day using a sliding window of 512 s. Both PSDs contain not only the (remaining) disturbances but also the ambient magnetic field. Their difference shows the absolute total power of the removed disturbances, while their ratio represents the minimum factor with which the power of the disturbances is reduced. The mean of this factor for the 24 h interval shown in Fig. 9 over the frequency range covering periods from 2 s to 1 min is equal to 7.8. For lower frequencies, in the range covering periods between 1 min to 6 h we obtain a factor of 3.9 from the PSDs computed without windowing.

The success of the cleaning procedure can also be estimated for each individual disturbance class. The initial magnitudes of the disturbances targeted for cleaning are shown in Table 2 for each sensor. Values are given for each component in the OSRF and for the module. The last column shows the remnants of the disturbances in the corrected data.

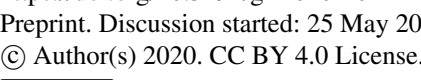



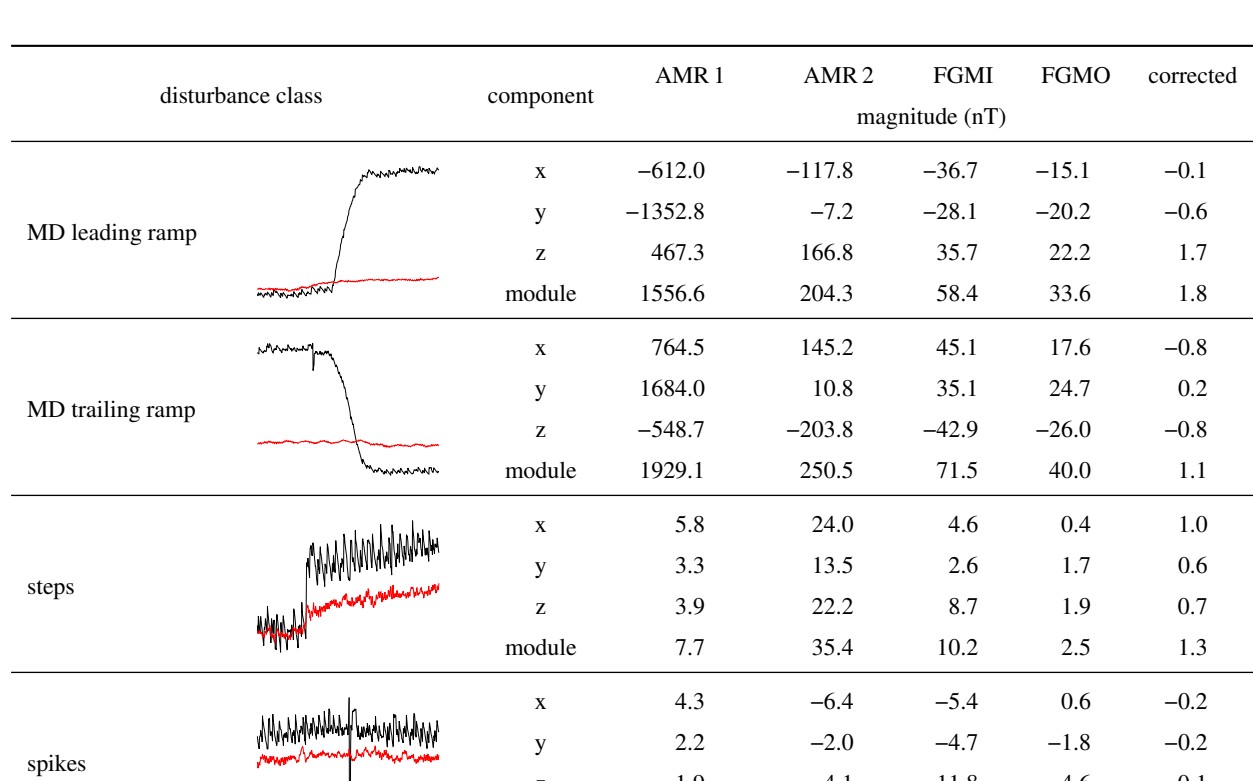

| disturbance class | | component | AMR 1 | AMR 2 | FGMI | FGMO | corrected |
|---|---|---|---|---|---|---|---|
| | | | magnitude (nT) | | | | |
| MD leading ramp | | x | −612.0 | −117.8 | −36.7 | −15.1 | −0.1 |
| | | y | −1352.8 | −7.2 | −28.1 | −20.2 | −0.6 |
| | | z | 467.3 | 166.8 | 35.7 | 22.2 | 1.7 |
| | | module | 1556.6 | 204.3 | 58.4 | 33.6 | 1.8 |
| MD trailing ramp | | x | 764.5 | 145.2 | 45.1 | 17.6 | −0.8 |
| | | y | 1684.0 | 10.8 | 35.1 | 24.7 | 0.2 |
| | | z | −548.7 | −203.8 | −42.9 | −26.0 | −0.8 |
| | | module | 1929.1 | 250.5 | 71.5 | 40.0 | 1.1 |
| steps | | x | 5.8 | 24.0 | 4.6 | 0.4 | 1.0 |
| | | y | 3.3 | 13.5 | 2.6 | 1.7 | 0.6 |
| | | z | 3.9 | 22.2 | 8.7 | 1.9 | 0.7 |
| | | module | 7.7 | 35.4 | 10.2 | 2.5 | 1.3 |
| spikes | | x | 4.3 | −6.4 | −5.4 | 0.6 | −0.2 |
| | | y | 2.2 | −2.0 | −4.7 | −1.8 | −0.2 |
| | | z | 1.9 | −4.1 | −11.8 | −4.6 | −0.1 |
| | | module | 5.2 | 7.9 | 13.8 | 4.9 | 0.3 |
| high frequency | | x | | 0.7 | 1.4 | 0.4 | 0.1 |
| | | y | | 0.1 | 1.1 | 0.3 | 0.0 |
| | | z | | 0.3 | 1.5 | 0.7 | 0.1 |
| | | module | | 0.8 | 2.3 | 0.9 | 0.1 |

**Table 2.** The initial magnitudes of the disturbances at all sensors and the final magnitudes in the corrected data for 2019.03.04. For the MD and for the spikes the sign shows the direction of the disturbance. AMR1 does not detect a quiet interval therefore we cannot estimate the HF disturbance magnitude at AMR1. The MD and the steps magnitudes are defined as the size of their ramps. The magnitudes of the spikes are equal with the spikes heights/depths. The high frequency disturbance magnitude is defined as the peak to peak amplitude. Samples of the disturbances affecting the $z$ component of the outboard sensor (black lines), together with the corrected measurements (red lines) over ten minute intervals are illustrated in the second column.





For the midnight disturbance we separated the leading ramp occurring around 15:00 UT from the abrupt trailing ramp about one hour later. The magnitude is computed as the difference between the median over 1.5 min of the field before and after the ramp. The leading ramp is reduced from about 34 nT in the FGMO measurements to less than 2 nT in the corrected measurements. The trailing ramp is reduced from 40 nT to about 1 nT. For the components, positive sign denotes upward ramp and negative sign downward ramp.

The ramps of the step-like disturbances are symmetric therefore we do not differentiate between the leading and the trailing ramps. The magnitudes are computed in the same way as for the MD. The mean step magnitude is reduced from 2.5 nT to 1.3 nT. However, note that the $x$ component is more than doubled, from 0.4 nT to 1 nT. This is a necessary compromise we have to make because the polarization directions of the disturbances are not orthogonal, as discussed in sec. 4.2.

The magnitude for the spikes was computed as the difference between the value of the peak of the spike and the median over 20 s intervals 5 s before and 5 s after the peak of the spike. For 2019.03.04 we obtain a mean magnitude of 13.8 nT for the initial FGMI measurements, 4.9 nT for the FGMO initial measurements, and 0.3 nT for the corrected measurements. For the components, positive sign denotes upward spikes and negative sign downward spikes.

To estimate the reduction of the high frequency disturbance we use as disturbance-free etalon the quiet 10 min interval visible in Fig. 4 between 00:40 and 00:50. The magnitude of the high frequency disturbance is computed as the difference between the mean peak to peak amplitude ( $2\sqrt{2\langle B^2\rangle_{\text{time}}}$ ) of the measurements during the reference quiet interval – which is 0.2 nT for the corrected measurements – and the mean peak to peak amplitude over the adjacent interval between the next two spikes. The result is below 0.1 nT for all components and for the module. Note that while AMR1 does not detect a quiet interval, it is still affected by a disturbance in the high frequency range of about 18 nT peak to peak amplitude, possibly coming from another source(s). Despite the large amplitude of this disturbance at AMR1, the increase of the disturbance in the high frequency range of the FGMO and FGMI measurements after the AMR correction is below 0.1 nT. All other discussed disturbances apart from the MD are lower at AMR1, which combined with the large scale factor used for correcting the MD assures minimum transfer of these disturbances to the corrected data.

## 6 Summary and conclusions

We propose a multi-sensor method for removing spacecraft generated disturbances from magnetic field data. The method employs principal component analysis to decouple multiple disturbance sources and minimize the introduction of artefacts to the components free of the targeted disturbance.

A pair of sensors can resolve up to three independent disturbers. While no prior knowledge on the disturber source is required, linear polarization of the disturbance is assumed, and the polarization direction of different disturbers should ideally be mutually orthogonal. The method is robust enough to provide sensible results even if these assumptions are not strictly met. There are however situations, such as non orthogonal disturbances from sources with large spatial separation compared with the distance to the sensors when two sensors are not enough to remove the disturbances with the described algorithm. Not linearly polarized disturbances, as those produced by reaction wheels, need special treatment not covered by this work.





We applied the PiCoG cleaning method to the GK2A SOSMAG sensor configuration by first using the spacecraft-body mounted AMR sensor measurements to remove large disturbances from the two boom mounted FGM sensors. Three distinct types of disturbances were then removed using the two FGM sensor measurements: high frequency disturbance in less than 1 min range, spikes occurring every 10 min, and steps occurring at intervals above 1 h.

We proved that on a specific day the method was able to reduce the spectral power of magnetic field disturbances by at least
a factor of 7.8 in the period range of 2 s to 1 min and 3.9 in the period range of 1 min to 6 h. These values are representative for the performance of the method over the entire 2019 year.

The final correction takes the form of a linear combination of the different sensor readings whose coefficients were determined. These coefficients were uploaded to the GK2A spacecraft, allowing for in-flight removal of spacecraft disturbances and near real-time delivery of cleaned magnetic field data. In future we shall apply the PiCoG method for post-processing of data
from other spacecraft, e.g. from BepiColombo (Benkhoff et al., 2010) and Cluster.

*Data availability.* SOSMAG data can be requested from the European Space Agency (ESA) and from the National Meteorological Satellite Center (NMSC) of the Korea Meteorological Administration (KMA)

*Author contributions.* All authors contributed equally to the manuscript.

*Competing interests.* The authors declare that they have no conflict of interest.

*Acknowledgements.* This work was financially supported by the Deutsches Zentrum für Luft- und Raumfahrt under contracts 50OC1803 and 50OC1403, by the General Support Technology Programme of ESA, contract 4000105630, and by the Space Situational Awareness (SSA) Programme of ESA, contract 4000117456.





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
