# Peer review of "Maximum Variance Gradiometer technique for removal of spacecraft-generated disturbances from magnetic field data"

_Geoscientific Instrumentation, Methods and Data Systems, 2020_

## Referee Comment (RC1) · Anonymous Referee #1 · 4 Jul 2020

This material is fully worth publishing as a working record of the cleaning of SOMAG magnetic field data. As an academic paper to discuss the technique which contributes to the better scientific results, I think, the authors have to revise it, at first, to distinguish the matter particular to the SOMAG case from the general matter.

Major comments :

1) The descriptions in section 2 should be considered, because they would be inadequate to explain the basics of the method proposed by the authors. The authors start with expressing the disturbances as the productions of dipole and quadrupole magnetic moments. However, the disturbance characteristic which makes the method described

in section 3 applicable is the linear independence at two sensor positions, and therefore disturbances are not necessary to be expressed by the magnetic moments. Although the magnetic moment model would be very useful to optimize the sensor positions and estimate the error, as author did in section 5 and Ness (1971) did, it is not essential to describe the principle of the method proposed by the authors.

2) Descriptions about general rule and requirements are mixed with those about specific conditions to SOMAG and author's assumptions. It makes the readers confuse what is universal to all magnetic field measurement with what is specific to author's case.

2-1) page 2 line 47, 'In many cases the direction of ...' I do not think it is often the case.

2-2) page 5 line 117, 'but does not change its direction' I do not think it is often the case.

2-3) page 6, line 157, 'For many spacecraft, including GK2A, artificial disturbances keep their direction fixed ...' I do not think it is often the case.

3) Many of equations in this paper are derived without enough explanation, and some of them seem to be incorrect.

3-1) page 3 lines from 70 to equation (8), this part is not understandable due to the shortage of the explanations.

3-2) page 3 line 73, what are k and l ?

3-3) page 3 line 79. 'The inverse ... ' Please explain the process to derive it. If $(3X-I)^{-1} = (3/2 X-I)$, as authors say, $(3X-I)(3/2X-I) = I$. The left is $9/2 X^2 - 9/2 X + I$, so it leads $X^2 = X$. Is it correct ?

3-4) page 4 line 84, 'and$(5X-2I)^{-1}$ is equal to $(5/6X-1/2I)$' if so, again, $X^2 = X$. Is it correct ?

3-5) page 7, line 196, 'To eliminate the disturbance ...' this sentence is difficult to

understand. Please make it easy to understand.

3-6) page 17, equations (29a)(29b)(29c), Please explain how these equations are derived.

3-7) page 17, lines 388-399, It is not clear what $\hat{G}$s expresses (it cannot be the absolute offset), and how equation (31) is derived.

Other specific comments

4) page 3 line 66, 'Because higher multipole moment ...' Here authors say that they can ignore the contribution by higher degree moments. However, later they discuss under the assumption that one of the sensor pair is very closely located to the disturbance source, and therefore the contribution by higher degree moments cannot be negligible. Please make the descriptions consistent.

5) The proposed strategy to remove the noise argued in this paper seems to be inconsistent. In page 6, line 151, 'We now assume that one of the terms in Eq. (9) is much larger than the others. ...' In page 10 line 2, 'the placement of the AMRs close to the disturbances sources.' To do it, the authors should know the positions of the disturbance sources to locate the sensors nearby. It is inconsistent with the advantage of this method, 'allows the separation of disturbances generated by the spacecraft ... without prior knowledge about the positions of the disturbances sources. (page 5, line 134)' Please make it consistent.

6) page 9, line 232, '3-axis Flux Gate Magnetometer (FGM)...' Is the outboard sensor built based on the design by Primdahl (1979) and inboard one is based on Acuna (2002) ? If not, please refer the papers more adequately.

7) page 10, lines 263-268, I suppose that the sensing alignment relationship between the FGM and AMR sensors would significantly affect the result of the removal of the magnetic disturbances. Please describe the knowledge about the alignment relationship and its accuracy.

8) As for the March 4 case presented in this paper, magnetic disturbances are caused by multiple sources and they can be discriminated because the repetition periods are very different one another. The authors should discuss the condition regarding the repetition periods of the disturbances when the proposed method works well and when it does not.

9) The order of Figure 2 and Figure 3 should be changed since Figure 3 appears earlier in the text.

10) The meaning of the word 'orthogonality' in this paper is not clear. If it means linear independence, 'up to three independent, mutually orthogonal, simultaneously active disturbances can be separated using two sensors. (page 5, line 118)' would not be correct. More than three disturbances may be separated if they are linearly independent. The statement in page 14, lines 323-332 should be revised.

11) Page 17, section 4.3, What is the advantage to remove the disturbances by the onboard processor ? Because it cannot be guaranteed that the coefficients do not change for long period, it would be much better to determine the coefficients from the raw data on the ground.

---

## Referee Comment (RC2) · Anonymous Referee #2 · 12 Aug 2020

Comments on **Principal Component Gradiometer technique for removal of spacecraft-generated disturbances from magnetic field data** by Ovidiu Dragos Constantinescu, Hans-Ulrich Auster, Magda Delva, Olaf Hillenmaier, Werner Magnes, and Ferdinand Plaschke.

The article presents an approach for correcting boom satellite magnetometer data by using reference magnetometers mounted within the satellite or on the boom closer to the satellite. The authors present the PiCoG algorithm (Principal Component Gradiometer algorithm). The idea is to identify the stray magnetic satellite field using a multitude of magnetometers and to remove it from the data.

The difference between measurements of different magnetometers show just the disturbing fields. The correction to be done depend linear on the measured differences. A Matrix **A** giving this linear dependence is to be determined and uploaded to the satellite for on board processing.

In **Section 2.1** the dipole field formula is used to show that a known dipole stray magnetic field at one magnetometer position, can be used to calculate the stray field at another magnetometer if positions of both magnetometers and of the disturbing dipole is known. The dipole moment is not needed. The same is done for a quadruple stray source.

In **Section 2.2** the situation with several sources is discussed: The contributions from more than one simultaneously active, arbitrary placed source with arbitrary time dependence cannot be separated from the ambient field in a simple way. At least two magnetometers are needed. Sources whose distance is small compared to the distance of the magnetometer act like a single multipole. It is stated: "For the procedure to work, the quadrupole contribution must be much weaker than the dipole contribution at both sensors."

Configurations are discussed that may simplify the evaluation: One sensor is placed close to a disturber allows eliminating this disturber. Three mutually orthogonal sources can be separated using two (three axe) Sensors by principal component analysis. If more than three sources are present more than two sensors are required.

In **Section 3** the PiCoG cleaning algorithm is illuminated. While the ambient magnet field is the same at all sensors, the disturbance is not and can therefore be removed. The difference of two sensors is independent of the ambient field (Eq. 9). The disturbance at a given sensor is related to the differences to all other

sensors by a Matrix A. The correction matrix A is to be computed directly from the measurements.

In **Section 3.1** it is assumed, that the disturbance to one sensor is much larger than to the others and an iterative procedure is proposed, dealing with one disturber first. To determine the direction of the stray field at each sensor, it is assumed that changes in the stray field are large compared to changes in the surrounding field values. The authors look at the variance of the measurements, identify the stray field variations and call the stray field direction at each sensor the "x-direction of a VPS (variance principal system)". The stray field measured at one sensor can be used to correct measurements at another sensor. The correction to the VPS-x direction of the i-the sensor is proportional to difference between measurements of j-th and i-th sensor. Constant of proportionality is the square of the quotient of variance of the disturbance to the i-sensor by the variance of the difference. This procedure is summed up in the Matrix **A** and gives the first estimate of **A** in the iterative process.

In **Section 3.1.1** the case of two sensors and the disturbing dipole being lined up is discussed.

In **Section 3.2** the successive use of further disturbers is formalized.

In **Section 4** an application to data of spacecraft GK2A is presented. Two magnetometers on a boom and two magnetometers at the platform are available.

What exactly has been done is illuminated in **Section 4.1 .** One of the platform magnetometers is not used due to noise considerations. The other platform magnetometer is used to remove disturbance at both boom magnetometers. A time period with a prominent disturbance is used to determine the respective correction matrices. The direction of the disturbing signal in the boom magnetometer system is determined by calculation the variance on a 1°x1° grid. A second-order correction using platform magnetometers was not done, once more for reasons of noise consideration.

**Remark:**

The presented method needs the disturbing sources to change with time (variance analysis). In spacecraft magnetical cleanliness DC magnetic disturbers play a big role. On the other hand the offset drift of fluxgate magnetometer is a

known problem. Is the method valid for DC calibration? Otherwise write "AC disturbance" in line 5 (Abstract) and in line 496 (Summary and Conclusion).

**Critics on the method:**

The authors report on their approach to correct measured magnetic data of the SOSMAG satellite. This is very interesting to coworkers on interpreting SOSMAG data. As reviewer I have to ask myself though, if the paper provides information useful beyond the SOSMAG satellite for a reader, who has the task of cleaning up magnetometer data. The authors call their method to deal with the SOSMAG data "PiCoG algorithm". It is more of a methodology than an algorithm that could be coded as is.

In chapter 2.1 interesting formulas are deduced for dipole and quadruple fields. They are used to show, that the magnetic field of a low frequency source can be factorized in a time-dependant and a geometry part. But is not that clear anyway for quasi DC magnetic sources? Do the formulas for the geometry factors enter the evaluation?

In chapter 3.1. it is assumed, that one of the magnetometers is very close to a disturber. Does the method also work, if that is not the case?

Unfortunately none of modern methods for analysis of multivariate time series is used. Principal component analysis is one of them. It uses spectral analysis of the cross-covariance matrix of all additional measured magnetometer components with respect to the reference magnetometer. The components of the first eigenvector (largest eigenvalue) directly delivers the set of direction consinus deduced in chapter 3.1 to define the "x-direction of the VPS" for each additional magnetometer. Of course a paper could be interesting if it goes beyond standard methods. But it should than be clear that it surpasses their results in a certain respect. The results of sec. 2.1, that means the known geometry factors, add information not been used in standard methods. Using for example the geometry factors for better identifying disturbing time series in the data, or for better determining the distribution of disturbing signal to the different magnetometers would render this paper interesting to a broader public.

The time dependence of the disturbing signal is not at all used in chapter 3, where the PiCoG Algorithme is defined. Nevertheless during the actual data evaluation the authors implicitly use the time dependence by looking at different

time periods, with different sources active. Fig. 2 shows the distribution of directions on a sliding window. Later on, in chapter 4. ramps and spikes are used to validate the result. These tricks should be included in the PiCoG algorithm.

The variance of measured date is used. That means field sources (ambient andmdisturbing fields) are understood as random processes. The motivation is not clear. In chapter 4.1 ramps and spikes in the measured time series are identified and used for validating the PiCoG results. The step amplitudes in all components could directly be used to deduce the geometry factors (Component of Matrix A in formula 10) between different magnetometers.

Fig. 4. Shows magnetometer values in the coordinate system orientated along the main axes of the data variances ellipsoid (VPS system). The figure shows, that the spike signal is still present in the z- and in the y-direction. Accordingly the VPS x-direction does not point along the spike disturbance.

**Further remarks or questions:**

L39: The PiCoG Process also is not running on the SC.

L48: This is the case if only variances are looked at. But the authors look later at ramps and spikes. They can even be identified, if they point along the ambient field.

L48: The term "principal component" is misleading. It usually refers to direction of a main axe of the stray ellipse in a multivariate random process.

L107: Perhaps better: "a collection of dipoles will in general generate multipole moments"

L 151: Which term is much larger? Would a strong disturber really make only one term large?

L137 this sentence would be more readable, if the summation symbol was omitted.

L157: Do you mean: "disturbing magnetic moments are fixed in direction with moments changing with time"?

L158: The stray field of one disturber has a constant direction in the magnetometer system. No need for a new coordinate system.

L166: Using this VPS suggests, that the disturber itself is a multivariate random process. But that is not the case. The VPS-x direction can be calculated by correlating the disturbing field strength with the measured x-, y- ,z- components. The term "variance principle system" is misleading. The reader could get the impression, that principal component analysis was done.

L167: Are the alpha i,j in Eq. 13a the same as the A i,j in Eq. 10? Than please use the same denomination.

L191: It is not clear to me if the b's are known at this point and if yes, where they are calculated.

L211: Do you mean "if stray fields of different disturbers are not coincident at the magnetometer location"?

I quit following the text here because the authors use a matrix notation, where I guess vectors of stray fields are sufficient.

**Conclusion:** The paper is an excellent report on how the authors achieved to clean and calibrate SOSMAG data. However the term "principal component technique" in the title is misleading. The authors should revise the method and try to use or at least refer to standard methods for multivariate data analysis and, if possible, expand them to produce a paper of more general interest.

---

## Author Comment (AC1) · 14 Aug 2020

We thank the Reviewer for the useful comments on our manuscript. We answer bellow to the points raised and we are going to change the manuscript accordingly.

*This material is fully worth publishing as a working record of the cleaning of SOMAG magnetic field data. As an academic paper to discuss the technique which contributes to the better scientific results, I think, the authors have to revise it, at first, to distinguish the matter particular to the SOMAG case from the general matter.*

*Major comments :*

[Figure]

*1) The descriptions in section 2 should be considered, because they would be inade-
quate to explain the basics of the method proposed by the authors. The authors start
with expressing the disturbances as the productions of dipole and quadrupole magnetic
moments. However, the disturbance characteristic which makes the method described
in section 3 applicable is the linear independence at two sensor positions, and therefore
disturbances are not necessary to be expressed by the magnetic moments. Although
the magnetic moment model would be very useful to optimize the sensor positions and
estimate the error, as author did in section 5 and Ness (1971) did, it is not essential to
describe the principle of the method proposed by the authors.*

**(1)** In section 2.1 we demonstrate that for single dipole/quadrupole disturbance sources
the problem of deriving the magnetic field produced by a disturber at one location us-
ing only the magnetic field measured at another location, has a solution and the so-
lution is unique (equations (2),(3) and (5),(6) on page 3). This allows expressing the
disturbance magnetic field as a linear combination of the difference between the mea-
surements (equation (7) on page 4, valid for both dipole and quadrupole disturbances).
This in turn is the justification for equation (10) in section 3 (see first sentence on page
6) on which the proposed cleaning algorithm is based. We believe that this justification
is essential for the proposed method and therefore the dipole/quadrupole description
of the disturbing sources is necessary.

*... the disturbance characteristic which makes the method described in section 3 ap-
plicable is the linear independence at two sensor positions ...*

On the contrary, equations (2) and (5) show that the disturbing magnetic field at one
position is a linear combination of the components of the disturbing magnetic field at
another position, therefore they are not linearly independent. The most general needed
characteristic for gradiometer-based methods is equation (7), i.e. the linear relation
between the contribution of the disturbance at one point in space and the difference
between the measurements taken at that point and those taken at another point. In

the particular case of our proposed PiCoG method, an additional necessary property is the linear polarization (i.e. one dimensional character) of the disturbance.

*2) Descriptions about general rule and requirements are mixed with those about specific conditions to SOMAG and author's assumptions. It makes the readers confuse what is universal to all magnetic field measurement with what is specific to author's case.*

**(2)**

We agree that the text is misleading. We have to distinguish between two types of disturbance sources. The first one ( the one we mentioned in the text) is caused by time variable currents. The field signature caused by this type of disturbance is identical at any measurement position and direction. Only the sign and amplitude depend on position and direction. One can rotate the field measured by a three axes sensor in a (principal axes) coordinate system in which the disturbance is present in one component only. This scalar type of disturbance needs 1 degree of freedom of a sensor difference signal for correction.

In contrast, a rotating magnet (at a sufficient large distance assumed as a rotating dipole) will produce a signature in two directions in a coordinate system aligned to the disturbance signal. We are not treating this type of disturbance in the present work.

We did our best to formulate the cleaning algorithm in its most general form. For single disturbance sources the most general approach is the gradiometer approach, expressed by equation (7). However, for the algorithm to work when multiple disturbers are present, there are several conditions to be met, which reduce the generality. The most important condition required by the PiCoG technique is that the disturbances to be cleaned vary only in magnitude and keep their direction constant. From the points (2.1), (2.2), (2.3) raised below, we conclude that by "specific conditions to SOMAG and author's assumptions" the Reviewer refers exactly to this constant direction re-
quirement. This condition is essential for the proposed algorithm. We state in the first paragraph of section 2.2 (lines 98-101) that the universal case of multiple arbitrary disturbers cannot be treated by the proposed algorithm. Next, on page 5 lines 121-122 we clearly state that "The PiCoG cleaning method assumes this type of linearly polarized disturbances". The next sentence allowing for "non-linearly polarized disturbances" is indeed an oversight on our part, confusing for the reader. We will remove this from the manuscript.

*2-1) page 2 line 47, 'In many cases the direction of ...' I do not think it is often the case.*

**(2.1)** We will change the text to make clear we refer to the disturbances due to time variable currents

*2-2) page 5 line 117, 'but does not change its direction' I do not think it is often the case.*

**(2.2)** We will change the text to make clear we refer to the disturbances due to time variable currents

*2-3) page 6, line 157, 'For many spacecraft, including GK2A, artificial disturbances keep their direction fixed ...' I do not think it is often the case.*

**(2.3)** We will change the text to make clear we refer to the disturbances due to time variable currents

*3) Many of equations in this paper are derived without enough explanation, and some of them seem to be incorrect.*

**(3)** Please see the answers (3.2) to (3.7)

*3-1) page 3 lines from 70 to equation (8), this part is not understandable due to the shortage of the explanations.*

[Figure]

**(3.1)** Please see the answers (3.2), (3.3) and (3.4)

*3-2) page 3 line 73, what are k and l ?*

**(3.2)** As stated in lines 73-74, subscripts stand for components, superscripts stand for positions. We will explain this more clear in the manuscript. $k$ and $l$ are the indices for the Cartesian components. $\hat{r}_k$ is the component $k$ of the unit vector $\hat{r}$.

*3-3) page 3 line 79. 'The inverse ... ' Please explain the process to derive it. If (3X-I)ˆ(-1) = (3/2 X-I), as authors say, (3X-I)(3/2X-I) = I. The left is 9/2 Xˆ2 - 9/2 X + I, so it leads Xˆ2 = X. Is it correct ?*

**(3.3)** We do indeed make use of the fact that $\mathcal{X}$ is an indempotent matrix, $\mathcal{X}^2 = \mathcal{X}$. On components, using the Einstein notation (summation over repeating indices):

$$(\mathcal{X}^2)_{ij} = X_{ik}X_{kj} = \hat{r}_i\hat{r}_k\hat{r}_k\hat{r}_j = \hat{r}_i|\hat{\boldsymbol{r}}|^2\hat{r}_j = \hat{r}_i\hat{r}_j = X_{ij} \tag{1}$$

Now we find $a$ and $b$ such as $(3\mathcal{X} - \mathcal{I})(a\mathcal{X} + b\mathcal{I}) = \mathcal{I}$

$$\Rightarrow (2a + 3b)\mathcal{X} - (1 + b)\mathcal{I} = 0 \quad \forall \mathcal{X} \quad \Rightarrow \quad a = 3/2 b = -1$$

Using the idempotency of $\mathcal{X}$ it is easy to check that indeed $(3\mathcal{X} - \mathcal{I})(3\mathcal{X}/2 - \mathcal{I}) = \mathcal{I}$

*3-4) page 4 line 84, 'and(5X-2I)ˆ(-1) is equal to (5/6X-1/2I)' if so, again, Xˆ2 = X. Is it correct ?*

**(3.4)** The inverse of $(5\mathcal{X} - 2\mathcal{I})$ is derived using a similar approach as detailed in (3.3). As proved above, $\mathcal{X}^2 = \mathcal{X}$.

*3-5) page 7, line 196, 'To eliminate the disturbance ...' this sentence is difficult to understand. Please make it easy to understand.*

[Figure]

**(3.5)** We change the formulation in the manuscript from
"To eliminate the disturbance $b_x^j$, the factor in front of it must vanish, therefore"
to
"Since the corrected magnetic field should be independent on the disturbing magnetic field $b_x^j$, results that the factors multiplying $b_x^j$ in Eqs. (18) must be zero, therefore"

*3-6) page 17, equations (29a)(29b)(29c), Please explain how these equations are derived*

**(3.6)** To derive the expressions for the matrices $\mathcal{M}$ in Eq. (28) we start by writing the third order correction of the AMR-corrected outboard sensor measurements ($B^{1,sa}$) using the AMR-corrected inboard sensor measurements ($B^{1,ta}$) as given by Eq. (26) ($i = s, \ j = t$):

$$B^{c,s} = B^{1,sa} + \mathcal{C}^s(B^{1,sa} - B^{1,ta})$$

with $\mathcal{C}^s$ given by Eq. (30) being the factor in front of $\Delta B^{0,ij}$ in Eq. (26). We then replace the first order AMR corrected inboard and outboard measurements $B^{1,sa}$ and $B^{1,ta}$ in the expression of $B^{c,s}$ above using Eqs. (27) and after some algebra we arrive at

$$B^{c,s} = \mathcal{M}^s B^{0,s} + \mathcal{M}^t B^{0,t} + \mathcal{M}^a B^{0,a}$$

with $\mathcal{M}^s, \mathcal{M}^t, \mathcal{M}^a$ given by Eqs. (29). Because the DC part of the disturbances is also removed, this form of $B^{c,s}$ does include implicitly the DC offset $G^s$ introduced by the correction.

*3-7) page 17, lines 388-399, It is not clear what Gˆs expresses (it cannot be the absolute offset), and how equation (31) is derived.*

**(3.7)** Eq. (31) is the definition of $G^s$. As the temporal average is taken over the entire time interval used to determine the correction matrices $\mathcal{M}$, $G^s$ represents the difference between the mean values before correction and the mean values after the correction, i.e. a constant offset between the original measurements of the outboard

sensor, $B^{0,s}$, and the corrected field. To reduce the correction to a purely AC correction we must subtract this offset from from the corrected field, hence the correction which does not introduce a DC offset (defined as the difference between the mean values before and after the correction) is given by Eq. (28). We will revise the formulation in the manuscript to avoid the confusion between the corrected measurements which include the DC offset change due to the AC correction and $B^{c,s}$ in Eq. (28) for which the DC offset change due to the AC correction is eliminated.

*4) page 3 line 66, 'Because higher multipole moment ...' Here authors say that they can ignore the contribution by higher degree moments. However, later they discuss under the assumption that one of the sensor pair is very closely located to the disturbance source, and therefore the contribution by higher degree moments cannot be negligible. Please make the descriptions consistent.*

**(4)**

While in theory one could place a sensor so close to a disturber such that the octopole (or higher) contribution becomes significant, it is difficult to imagine a real life scenario where the dipole and quadrupole contributions of the disturber do not vastly overwhelm the octopole (or higher) contribution. We are indeed assuming in section 3.1 lines 151-153 that the distance from one sensor to the disturber being currently cleaned is small *relative* to the distance to the other disturbers. This does not imply a distance small enough to make octopole and higher orders visible. It merely means that the disturbers are not equally distanced from the sensor in question. On page 4 lines 106-110 we state that the dipole and quadrupole contributions should not have comparable strengths at the sensor location. We will change this sentence to clarify that also higher multipoles should be weak compared with the dominant dipole or quadrupole being cleaned.

*5) The proposed strategy to remove the noise argued in this paper seems to be in-*

*consistent. In page 6, line 151, 'We now assume that one of the terms in Eq. (9) is much larger than the others. ...' In page 10 line 2, 'the placement of the AMRs close to the disturbances sources.' To do it, the authors should know the positions of the disturbance sources to locate the sensors nearby. It is inconsistent with the advantage of this method, 'allows the separation of disturbances generated by the spacecraft ... without prior knowledge about the positions of the disturbances sources. (page 5, line 134)' Please make it consistent.*

**(5)**

The observation is correct, of course some knowledge about the disturbers positions is necessary. For instance if a disturber is placed at equal distances from two sensors, the proposed procedure using those two sensors cannot work. It is also assumed that the boom-tip placed sensor is further away from disturbers than the other sensors, and that when the body-mounted sensors accommodations were decided at least some minimum information about the locations of major disturbers was available so sensors could be placed in their vicinity. However, apart from that, the positions of the disturbers do not enter in any way in the cleaning procedure, hence the statement in line 134. We will clarify this in the manuscript.

***6)** page 9, line 232, '3-axis Flux Gate Magnetometer (FGM)...' Is the outboard sensor built based on the design by Primdahl (1979) and inboard one is based on Acuna (2002) ? If not, please refer the papers more adequately.*

**(6)**

Both sensors are neither designed similar to the sensors described in the early Acuna or in the Primdahl papers. The reference refers to the fluxgate principle only. The Mario Acuna design consist of three single component sensors accommodated next to each other. The disadvantage of these sensors is that the axes directions are determined by the ringcores and the pickup windings and thus they are not stabilised by the more

robust feedback coils. At low field conditions (e.g. for the Voyager spacecraft) this is not of importance, however for e.g. MAGSAT it causes a significant uncertainty. Fritz Primdahl has therefore developed a vector compensated sensor in which a sensor similar to the sensors developed by Acuna has been placed in a sophisticated feedback coil system compensating the field for all three single sensors in all directions. Xavier Lalanne developed a very nice sensor in which 6 ringcores are placed on the sides of a cube inside a Helmholtz coil system. This is a great design because it is fully symmetric, however very elaborated. Our design is based on two crossed ringcores in the centre of a Helmholtz system. It is described in detail in the Themis Magnetometer paper by Auster at al. which has been added to the list of references.

*7) page 10, lines 263-268, I suppose that the sensing alignment relationship between the FGM and AMR sensors would significantly affect the result of the removal of the magnetic disturbances. Please describe the knowledge about the alignment relationship and its accuracy.*

**(7)** Actually the alignment between the FGM and AMR sensors plays no role in the PiCoG method. This is because the cleaning is performed only on the maximum variance components of the measurements which are independent on alignment.

*8) As for the March 4 case presented in this paper, magnetic disturbances are caused by multiple sources and they can be discriminated because the repetition periods are very different one another. The authors should discuss the condition regarding the repetition periods of the disturbances when the proposed method works well and when it does not.*

**(8)**

That is correct. The proposed method works well when the polarization direction of the targeted disturbance is determined by the maximum variance direction. If the disturbances are in the same frequency range and therefore cannot be decoupled by using
different window lengths one must either find their polarization direction using other means, or they must have magnitudes different enough such that the dominant disturbance – being currently cleaned – determines the maximum variance direction. We will introduce a paragraph clarifying this in section 5.

*9) The order of Figure 2 and Figure 3 should be changed since Figure 3 appears earlier in the text.*

**(9)** This is correct, it will be fixed in the manuscript.

*10) The meaning of the word 'orthogonality' in this paper is not clear. If it means linear independence, 'up to three independent, mutually orthogonal, simultaneously active disturbances can be separated using two sensors. (page 5, line 118)' would not be correct. More than three disturbances may be separated if they are linearly independent. The statement in page 14, lines 323-332 should be revised.*

**(10)** In the manuscript, the word "orthogonal" has the common geometrical meaning: Two directions are called orthogonal if they form a right angle. We say that two disturbances are orthogonal if their maximum variance directions are orthogonal to each other. We will explain this better in the manuscript.

*11) Page 17, section 4.3, What is the advantage to remove the disturbances by the onboard processor ? Because it cannot be guaranteed that the coefficients do not change for long period, it would be much better to determine the coefficients from the raw data on the ground.*

**(11)** The coefficients were determined from raw data on the ground. Monitoring the magnetic field at geostationary orbit supplies important information about the space weather events reaching the Earth. On-board data cleaning provides near real-time accurate magnetic field data which is essential in this context. An added benefit is a four fold increase in the time resolution achieved by changing the telemetry from raw
data from four sensors at one vector per second to cleaned data at four vectors per second. We will include this information in the manuscript.

---

## Author Comment (AC2) · 21 Aug 2020

We thank the Reviewer for the constructive remarks and critics.

*... The presented method needs the disturbing sources to change with time (variance analysis). In spacecraft magnetical cleanliness DC magnetic disturbers play a big role. On the other hand the offset drift of fluxgate magnetometer is a known problem. Is the method valid for DC calibration? Otherwise write "AC disturbance" in line 5 (Abstract) and in line 496 (Summary and Conclusion). ...*

• If the mean field produced by a disturber is different from zero, i.e. there is a non-zero

[Figure]

DC disturbance due to the targeted disturber – which is most of the times the case, the proposed method will automatically correct this DC disturbance if it depends in the same way on the distance to the source as the cleaned AC term (dipole/quadrupole). The total DC shift introduced by the final correction is contained in the $G^s$ vector given by equation (31). However, even if the above condition is true, the method presented here does not provide the DC offset produced by completely time independent disturbers, therefore it cannot be used for the DC correction. Moreover, the internal offset drift of the sensors cannot be treated using the PiCoG technique. We will make this clear in the Abstract and in the Conclusions.

*... The authors call their method to deal with the SOSMAG data "PiCoG algorithm". It is more of a methodology than an algorithm that could be coded as is.*

• That is correct. We abused the word "algorithm". We will replace "algorithm" with "technique/method" throughout the manuscript.

*In chapter 2.1 interesting formulas are deduced for dipole and quadruple fields. They are used to show, that the magnetic field of a low frequency source can be factorized in a time-dependant and a geometry part. But is not that clear anyway for quasi DC magnetic sources?*

• Indeed, equations (1) and (4), which are just the expressions for the magnetic field produced by a dipole/quadrupole show the trivial fact that in this case the space dependence can be separated by the time dependence as a factor. The key relations here are equations (2) and (5) which show that the magnetic field produced by a time-dependent dipole/quadrupole at a given location can be obtained using a *time independent* transformation of the magnetic field measured at a different location. This proves that it is in principle possible to use the measurements from one sensor to correct the measurements delivered by another sensor at different location. This is the foundation of our approach.

*Do the formulas for the geometry factors enter the evaluation?*

• No, the $\mathcal{T}$ matrices given by equations (3) and (6) are not used as such by the PiCoG technique. They might be used perhaps in a very controlled environment when the positions of the sensors and disturbers, as well as the decay law (dipole/quadrupole) of the disturber are precisely known. This work's goal is to provide a technique which does not require this information. However, the correction matrix $\mathcal{A}$ is related to the $\mathcal{T}$ matrix by the relation in line 146. This means that – once the $\mathcal{A}$ is computed using the PiCoG method – one could derive the corresponding $\mathcal{T}$ matrix. This is beyond the scope of the present work.

*In chapter 3.1. it is assumed, that one of the magnetometers is very close to a disturber. Does the method also work, if that is not the case?*

• This depends on the specific sensors-disturbers configuration. E.g. if more disturbers of the same type occupy a volume which is small compared to the distance to the closest sensor, the method works even if the sensor is not closer to one of the disturbers than to the others. If the directions of maximum variance of two disturbers are orthogonal to each other, again the method works even if the strength of the two disturbances are the same / disturbers placed at similar distances to the sensor. There are however situations in which the method does not work, as detailed in section 5.

*Unfortunately none of modern methods for analysis of multivariate time series is used. Principal component analysis is one of them. It uses spectral analysis of the cross-covariance matrix of all additional measured magnetometer components with respect to the reference magnetometer. ...*

• We are not sure we understand the point raised by the Reviewer. We *do* use Principal Component Analysis (PCA) as a key method employed by PiCoG. We state this e.g. in lines 119-120, 164-166, 496-497. We indeed determine the direction of maximum variance from the eigenvectors of the covariance matrix as described e.g. in section

1.4 of *Time Series Data Analyses in Space Physics, Song and Russell, SSR (1999)* or in *Analysis methods for multi-spacecraft data, Sonnerup and Scheible, ISSI Sci. Rep. SR-001, p185-220, Ed Paschmann and Daly, (1998)*. We do not perform a spectral analysis though because it was not necessary for our specific problem. Of course, if one knows – or determines – in advance that the disturbance to be removed is confined in a specific frequency band, one may perform the PCA in the frequency domain and select the eigenvectors corresponding to the frequency band of the disturbance. We will comment on this possibility in the revised text.

*... The results of sec. 2.1, that means the known geometry factors, add information not been used in standard methods. Using for example the geometry factors for better identifying disturbing time series in the data, or for better determining the distribution of disturbing signal to the different magnetometers would render this paper interesting to a broader public.*

• It is true that in this work we did not directly exploit equations (3) and (6) which give the exact expression of the "propagator" matrix $\mathcal{T}$ which allows computing the disturbance at one point in space once the disturbance at another point is known. This would allow cleaning the disturbances using a precise model representing the disturbance sources and the sensors positions. However, this is not the goal of the present work. Here we determine the correction matrices $\mathcal{A}$ – which are equivalent with the $\mathcal{T}$ matrices – solely from the available measurements. Of course, it might be possible to develop an entirely different method using the $\mathcal{T}$ matrices. However, making more intensive use of the $\mathcal{T}$ matrices is not necessary in the context of the present work.

*The time dependence of the disturbing signal is not at all used in chapter 3, where the PiCoG Algorithme is defined.*

• The time dependence is implicitly used through the fact that the correction is applied to the principal variance component.
*... Nevertheless during the actual data evaluation the authors implicitly use the time dependence by looking at different time periods, with different sources active. Fig. 2 shows the distribution of directions on a sliding window. Later on, in chapter 4. ramps and spikes are used to validate the result. These tricks should be included in the PiCoG algorithm.*

• We made efforts to keep the PiCoG method described in section 3 as general as possible. Including procedures specific to our particular application of the method to the SOSMAG data would in our opinion induce confusion to the reader. Presenting these procedures in the application section on the other hand, lets the reader decide for him/herself if these procedures are appropriate or not for his/her problem at hand. Even for our specific problem we did not used the same procedures from the beginning to the end: For the AMR correction we determined the maximum variance direction using just one step-like disturbance, while for the FGM-FGM correction we decided for a statistical approach using a sliding window.

*The variance of measured date is used. That means field sources (ambient andmdis- turbing fields) are understood as random processes. The motivation is not clear.*

• It is true that variance normally refers to the the deviation of a random variable from its mean value. A certain randomness is introduced by the ambient field. However, in the context of the present work, the random/non-random character of the disturbance plays no role. We use the variance only as a measure of how strong the AC disturbance is in each direction, and through PCA we determine the direction in which the variance is largest therefore the disturbance is strongest.

*...The step amplitudes in all components could directly be used to deduce the geometry factors (Component of Matrix A in formula 10) between different magnetometers.*

• The correction matrix $A$ is composed from a rotation and a scaling. We don't see a direct way to deduce the $A$ matrix from step amplitudes. After the rotation in the VPS

one could indeed determine the amplitude of the steps as we did in section 5 and from them derive the $\alpha$ factor in equation (13a). We think however that equation (14) gives a more general solution. Both estimates of the $\alpha$ factor are susceptible to improvements anyway as mentioned in lines 273-274.

*Fig. 4. Shows magnetometer values in the coordinate system orientated along the main axes of the data variances ellipsoid (VPS system). The figure shows, that the spike signal is still present in the z- and in the y-direction. Accordingly the VPS x-direction does not point along the spike disturbance.*

• This is correct. The $x$-axis of the VPS in Figure 4 is aligned with the variance direction of the highest frequency disturbance (first to be cleaned), distinct from the direction of the spikes. The VPS in which the data in Figure 5 is represented has its $x$-axis aligned with the direction of the spikes.

*L39: The PiCoG Process also is not running on the SC.*

• The PiCoG technique delivers the correction matrices $\mathcal{M}$ which are uploaded to the spacecraft and used for onboard data cleaning. As far as we understand, the *Poppe et al. (2011)* procedure cannot be reduced to a simple linear combination which can easily be implemented onboard.

*L48: This is the case if only variances are looked at. But the authors look later at ramps and spikes. They can even be identified, if they point along the ambient field.*

• We use the PCA also for the ramps and for the spikes. One could use other methods to determine the direction of the rams or spikes disturbances (even manually perhaps), but the PCA delivers the correct directions and the scale factors in an automatic fashion. We will emphasize in the text the use of PCA for spikes and ramps.

*L48: The term "principal component" is misleading. It usually refers to direction of a main axe of the stray ellipse in a multivariate random process.*
• The term "principal component" in the text does indeed refer to the main axis of the variance ellipsoid. The fact that the disturbance is not a random process does not affect neither the application of the PCA nor its results. We will comment in the text on the non-randomness of the the disturbances in order to prevent possible confusion.

*L107: Perhaps better: "a collection of dipoles will in general generate multipole moments"*

• The suggested formulation is indeed better. Thank you, we will use it in the text.

*L 151: Which term is much larger? Would a strong disturber really make only one term large?*

• Since the summation index $q$ in equation (9) refers to the disturbance sources – as detailed in line 137, the term corresponding to the strong/close disturber will be larger. A strong disturber will only affect the corresponding term in the sum.

*L137 this sentence would be more readable, if the summation symbol was omitted.*

• We will re-phrase this sentence to make it more readable.

*L157: Do you mean: "disturbing magnetic moments are fixed in direction with moments changing with time"?*

• Yes, this is what we mean. We will adapt the text to follow the Reviewer's suggestion.

*L158: The stray field of one disturber has a constant direction in the magnetometer system. No need for a new coordinate system.*

• The stray field of one disturber has indeed a constant direction in the magnetometer system. However, a new coordinate system is needed to align this direction with one of the coordinate system axes (in our case the $x$-axis).

*L166: Using this VPS suggests, that the disturber itself is a multivariate random process. But that is not the case. The VPS-x direction can be calculated by correlating the disturbing field strength with the measured x-, y- ,z- components. The term "variance principle system" is misleading. The reader could get the impression, that principal component analysis was done.*

• PCA was in fact done in order to obtain the disturbance direction. As stated before, the non-random character of the disturbing field is not relevant in this context. One could probably obtain the disturbance direction by minimizing the correlation between the disturbing field strength and the measured $y$ and $z$ components using as free parameters the angles of rotation for the new system. We do not see the advantage in using this alternative method.

*L167: Are the alpha i,j in Eq. 13a the same as the A i,j in Eq. 10? Than please use the same denomination.*

• They are not the same. $\mathcal{A}^{ij}$ in equation (10) is the matrix used to correct sensor $i$ measurements using sensor $j$ measurements. $\alpha^{0,ij}$ in equation (13) is a scalar scale factor given by equation (14) for the first order correction.

*L191: It is not clear to me if the b's are known at this point and if yes, where they are calculated.*

• The disturbance $b$ at the sensor position is not known at this point. We only make use of the dependence on the distance of $b$ as stated in lines 193-195.

*L211: Do you mean "if stray fields of different disturbers are not coincident at the magnetometer location"?*

• We mean "if stray fields of different disturbers do not share the same direction at the magnetometer location". We will adapt the text.

*I quit following the text here because the authors use a matrix notation, where I guess vectors of stray fields are sufficient.*

• We do not see how to concisely write the relations without using matrix notation.

*Conclusion: The paper is an excellent report on how the authors achieved to clean and calibrate SOSMAG data. However the term "principal component technique" in the title is misleading. The authors should revise the method and try to use or at least refer to standard methods for multivariate data analysis and, if possible, expand them to produce a paper of more general interest.*

• We thank the Reviewer for the appreciative comment. However, as explained above on several occasions, PiCoG *is* using principal component analysis as a essential tool, therefore we believe the title is appropriate. We will revise the paper nevertheless to emphasise the use of this standard analysis method.

---

## Referee Comment (RC3) · Anonymous Referee #2 · 1 Sep 2020

**Gi_2020-10 R2, Reaction to responses**

**"PiCoG algorithm":** Would you think that the following procedure is in line with what you did. Could this be a line out of an algorithm?

1. Define one of the N instruments as "reference instrument".
2. Calculate the differences of all instruments and the reference instrument.
3. In all difference signals, identify the instrument j showing the strongest disturbance.
4. Use this difference to correct all instrument readings using Eg. 13
5. For the next iteration start over with 2. disregarding instrument j.

**Remarks:** The number of iterations is restricted to N-1 (N the number of instruments).

Instead of a statistical method, step 4. could also be done using the scalar product of the difference signal and the signal to be corrected (correlation).

The method reminds to the Gram-Schmidt process for orthogonalizing a set of vectors. These vectors being time series of magnetometer data. Their dimension is the number of samples.

The result of the algorithm is certainly not the same if different time periods of data are used. This depends on the intensity of disturbers switched on during the considered time span. Here PCA and factor analysis could be used to identify time series of disturbing signals.

**PCA:** You determine the main axes in the 3D distribution of magnetometer measurements and in the distribution of differences between two different magnetometers. Then you assume that your $\alpha$ can be calculated based on the quotient of variances along these main axes (Eq. 14). This is a very bold assumption and you named quite some requirement for this assumption. Asking for PCA, I meant to use PCA in the 3N dimensions of all available measured time series. If PCA is referred to in the title the reader will expect it to be used on the multivariate time series $(X_1(t), Y_1(t), Z_1(t), \Delta X_{21}(t), \Delta Y_{21}(t), \Delta Z_{21}(t), \Delta X_{31}(t), \Delta Y_{31}(t), \Delta Z_{31}(t),\ldots)$. "1" being the reference magnetometer. This automatically produces what you call VPS-x directions (as components of the largest eigenvector). Mentioning PCA in this respect in the title, one would certainly also expect a factor analysis. That means an estimate of the time series of the disturbing signals by projection of the data vector to the eigenvector

directions. My use of the term "spectral analysis" was perhaps misleading. I meant the spectrum of eigenvalues of the crosscovariance matrix of all measured data time series (https://en.wikipedia.org/wiki/Factor_analysis). This would also reveal the number of relevant disturbers as the number of eigenvelues essentially differing from zero. It also quantifies the content of disturbing signal in the reference magnetometer readings. Therefore it would be a straight forward method to calculate the $\alpha$ values.

Along that you get a measure for the correlations of differences and disturbance at sensor 1. Therefore my question: "In chapter 3.1. is assumed, that one of the magnetometers is very close to a disturber. Does the method also work, if that is not the case?" If a disturbing signal is present in the Deltas even with small intensities it can easily be identified by PCA and factor analysis for removal. Estimates of corrections of higher order (disturbers 2, 3, ...) result as projections on eigenvalues that are next smaller than the first.

Please judge for yourself whether the reference to PCA in the title is really justified.

But even PCA and factor analysis do not deliver unique results. In PCA geometry factors are completely ignored. Therefore exploitation of Eg. 3 and Eq. 6 would introduce a completely new idea going further than what can be done by PCA.

On page 6 between line 157 and line 180 you argue very intuitively. This lack of mathematical rigor should be mitigated using PCA in the way I proposed.

It is absolutely not clear how Eq. 10 follows from Eq. 9. I even doubt, that a linear relation between the correction value for $B^i$ and the $\Delta B^{i,j}$ exists. This is only true if only one single disturber is on. I guess Eq.10 is the first order approach assuming that a certain disturber is very prominent (at a certain time span) in the difference $\Delta B^{i,j}$. Please clarify and explain that in the text.

---

## Author Response (AR1)

**Principal Component Gradiometer technique for removal of**
**spacecraft-generated disturbances from magnetic field data**
**– Response to the Reviewers –**

We thank the Referees for helping us improve the manuscript, which we revised according to their suggestions. In addition, we revised section 2.1 to correct the expression of the matrix $\mathcal{G}^{ij}$ entering the quadrupole correction, which we mistakenly took to be equal to the rotation matrix which transforms the versor $\hat{\boldsymbol{r}}^j$ to the versor $\hat{\boldsymbol{r}}^i$. The correct expression is: $\mathcal{G}^{ji} = \mathcal{Q}\mathcal{R}^{ij}\mathcal{Q}^{-1}$ with $\mathcal{R}^{ij}$ being the rotation matrix and $\mathcal{Q}(t)$ the quadrupole moment. This change has no implication on our treatment if only the magnitude of the quadrupolar disturbance changes with time. This was anyway needed by our proposed cleaning method. We explained this in the manuscript (last paragraph of section 2.1).

Below is our response to the Referees comments. It closely follows the Author Comments submitted in the Public Discussion. The Referee comments are typeset in italics, our answers are marked with the • symbol, and the descriptions of the changes made to the manuscript are marked with the ⋆ symbol. A version of the revised manuscript with tracked changes has been also provided.

**Referee #1 comments**

*This material is fully worth publishing as a working record of the cleaning of SOMAG magnetic field data. As an academic paper to discuss the technique which contributes to the better scientific results, I think, the authors have to revise it, at first, to distinguish the matter particular to the SOMAG case from the general matter.*

*Major comments :*

*1) The descriptions in section 2 should be considered, because they would be inadequate to explain the basics of the method proposed by the authors. The authors start with expressing the disturbances as the productions of dipole and quadrupole magnetic moments. However, the disturbance characteristic which makes the method described in section 3 applicable is the linear independence at two sensor positions, and therefore disturbances are not necessary to be expressed by the magnetic moments. Although the magnetic moment model would be very useful to optimize the sensor positions and estimate the error, as author did in section 5 and Ness (1971) did, it is not essential to describe the principle of the method proposed by the authors.*

• In section 2.1 we demonstrate that for single dipole/quadrupole disturbance sources the problem of deriving the magnetic field produced by a disturber at one location using only the magnetic field measured at another location, has a solution and the solution is unique (equations (2),(3) and (5),(6) on page 3 of the original manuscript ([OM]) / equations (2),(3) and (6),(7) on page 3-4 of the revised manuscript ([RM]) ). This allows expressing the disturbance magnetic field as a linear combination with time independent coefficients of the difference between the measurements (equation (7) on page 4[OM]/

eq.4 p.4[RM], valid for both dipole and quadrupole disturbances). This in turn is the justification for equation (10) in section 3 (see lines 146-148[OM]/166-168[RM]) on which the proposed cleaning algorithm is based. We believe that this justification is essential for the proposed method and therefore the dipole/quadrupole description of the disturbing sources is necessary.

*... the disturbance characteristic which makes the method described in section 3 applicable is the linear independence at two sensor positions ...*

• On the contrary, equations (2) and (5)[OM]/(2),(6)[RM] show that the disturbing magnetic field at one position is a linear combination of the components of the disturbing magnetic field at another position, therefore they are not linearly independent. The most general needed characteristic for gradiometer-based methods is equation (7)[OM]/(4)[RM], i.e. the linear relation between the contribution of the disturbance at one point in space and the difference between the measurements taken at that point and those taken at another point. In the particular case of our proposed PiCoG method, an additional necessary property is the linear polarization (i.e. one dimensional character) of the disturbance.

⋆ If the reviewer actually meant "dependence" (typo error) then we fully agree. We added a few sentences clarifying this at the end of section 2.1 (line 96[OM]/101-102[RM])

*2) Descriptions about general rule and requirements are mixed with those about specific conditions to SOMAG and authors assumptions. It makes the readers confuse what is universal to all magnetic field measurement with what is specific to authors case.*

• We agree that the text can be misleading. We have to distinguish between two types of disturbance sources. The first one (the one we mentioned in the text) is caused by time variable currents. The field signature caused by this type of disturbance is identical at any measurement position and direction. Only the sign and amplitude depend on position and direction. One can rotate the field measured by a three axes sensor in a (principal axes) coordinate system in which the disturbance is present in one component only. This scalar type of disturbance needs 1 degree of freedom of a sensor difference signal for correction.

In contrast, a rotating magnet (at a sufficient large distance assumed as a rotating dipole) will produce a signature in two directions in a coordinate system aligned to the disturbance signal. We are not treating this type of disturbance in the present work.

We did our best to formulate the cleaning algorithm in its most general form. For single disturbance sources the most general approach is the gradiometer approach, expressed by equation (7)[OM]/(4)[RM]. However, for the algorithm to work when multiple disturbers are present, there are several conditions to be met, which reduce the generality. One important condition required by the PiCoG technique is that the disturbances to be cleaned should vary only in magnitude and should keep their direction constant. From the points (2.1), (2.2), (2.3) raised below, we conclude that by "specific conditions to SOMAG and authors assumptions" the Reviewer refers specifically to this constant direction requirement. This condition is essential for the proposed algorithm. We state in the first paragraph of section 2.2 (lines 98-101[OM]/117-120[RM]) that the universal case of multiple arbitrary disturbers cannot be treated by the proposed algorithm. Next, on page 5 lines 121-122[OM]/141-142[RM] we clearly state that "The PiCoG cleaning method assumes this type of linearly polarized disturbances". The next sentence allowing for "non-linearly polarized disturbances" is indeed an oversight on our part, confusing for the reader.

⋆ We added several sentences explaining that the proposed cleaning method only deals with disturbances for which only the module (and not the direction) of the magnetic field changes (after line 49[OM]/54-57[RM]).

⋆ We removed "or non-linearly polarized disturbances" from line 122[OM].

**2-1)** *page 2 line 47, In many cases the direction of ... I do not think it is often the case.*

⋆ We changed the text after line 47[OM]/51-54[RM] to specify that we refer to disturbances which are large compared to the ambient field during the interval used for cleaning.

**2-2)** *page 5 line 117, but does not change its direction I do not think it is often the case.*

• It is explained on lines 120-122[OM]/140-142[RM] that we refer to the disturbances due to time variable currents

**2-3)** *page 6, line 157, For many spacecraft, including GK2A, artificial disturbances keep their direction fixed ... I do not think it is often the case.*

⋆ We changed the text starting with line 157[OM]/186-189[RM] to make it clear that we refer to disturbances due to time variable currents.

**3)** *Many of equations in this paper are derived without enough explanation, and some of them seem to be incorrect.*

• Please see the answers (3.2) to (3.7)

**3-1)** *page 3 lines from 70 to equation (8), this part is not understandable due to the shortage of the explanations.*

• Please see the answers (3.2), (3.3) and (3.4)

**3-2)** *page 3 line 73, what are k and l ?*

• As stated in lines 73-74[OM]/82-83[RM], subscripts stand for components, superscripts stand for positions. $k$ and $l$ are the indices for the Cartesian components. $\hat{r}_k$ is the component $k$ of the unit vector $\hat{\boldsymbol{r}}$.

⋆ We explained the meaning of the subscripts and of the superscripts more clearly in the text (lines 74[OM]/83[RM] and 77[OM]/86[RM]).

**3-3)** *page 3 line 79. The inverse ... Please explain the process to derive it. If (3X-I)ˆ(-*

*1) = (3/2 X-I), as authors say, (3X-I)(3/2X-I) = I. The left is 9/2 X^2 - 9/2 X + I, so it leads X^2 = X. Is it correct ?*

• We do indeed make use of the fact that $\mathcal{X}$ is an indempotent matrix, $\mathcal{X}^2 = \mathcal{X}$. On components, using the Einstein notation (summation over repeating indices):

$$\left(\mathcal{X}^2\right)_{ij} = X_{ik}X_{kj} = \hat{r}_i\hat{r}_k\hat{r}_k\hat{r}_j = \hat{r}_i|\hat{\boldsymbol{r}}|^2\hat{r}_j = \hat{r}_i\hat{r}_j = X_{ij} \tag{1}$$

Now we find $a$ and $b$ such as $\left(3\mathcal{X} - \mathcal{I}\right)\left(a\mathcal{X} + b\mathcal{I}\right) = \mathcal{I}$

$$\Rightarrow (2a + 3b)\mathcal{X} - (1 + b)\mathcal{I} = 0 \quad \forall \mathcal{X} \quad \Rightarrow \quad \begin{matrix} a = 3/2 \\ \text{and} \\ b = -1 \end{matrix}$$

Using the idempotency of $\mathcal{X}$ it is easy to check that indeed $(3\mathcal{X} - \mathcal{I})(3\mathcal{X}/2 - \mathcal{I}) = \mathcal{I}$

⋆ For the sake of readability, we do not include these details in the manuscript.

***3-4)*** *page 4 line 84, and(5X-2I)^(-1) is equal to (5/6X-1/2I) if so, again, X^2 = X. Is it correct ?*

• The inverse of $(5\mathcal{X} - 2\mathcal{I})$ is derived using a similar approach as detailed in (3.3). As proved above, $\mathcal{X}^2 = \mathcal{X}$.

***3-5)*** *page 7, line 196, To eliminate the disturbance ... this sentence is difficult to understand. Please make it easy to understand.*

⋆ We change the formulation in the manuscript from
"To eliminate the disturbance $b_x^j$, the factor in front of it must vanish, therefore"
to
"Since the corrected magnetic field should be independent on the disturbing magnetic field $b_x^j$, results that the factors multiplying $b_x^j$ in Eqs. (18) must be zero, therefore"

***3-6)*** *page 17, equations (29a)(29b)(29c), Please explain how these equations are derived*

• To derive the expressions for the matrices $\mathcal{M}$ in Eq. (28) we start by writing the third order correction of the AMR-corrected outboard sensor measurements ($\boldsymbol{B}^{1,sa}$) using the AMR-corrected inboard sensor measurements ($\boldsymbol{B}^{1,ta}$) as given by Eq. (26) ($i = s$, $j = t$):

$$\boldsymbol{B}^{c,s} = \boldsymbol{B}^{1,sa} + \mathcal{C}^s\left(\boldsymbol{B}^{1,sa} - \boldsymbol{B}^{1,ta}\right)$$

with $\mathcal{C}^s$ given by Eq. (30) being the factor in front of $\Delta\boldsymbol{B}^{0,ij}$ in Eq. (26). We then replace the first order AMR corrected inboard and outboard measurements $\boldsymbol{B}^{1,sa}$ and $\boldsymbol{B}^{1,ta}$ in the expression of $\boldsymbol{B}^{c,s}$ above using Eqs. (27) and after some algebra we arrive at

$$\boldsymbol{B}^{c,s} = \mathcal{M}^s\boldsymbol{B}^{0,s} + \mathcal{M}^t\boldsymbol{B}^{0,t} + \mathcal{M}^a\boldsymbol{B}^{0,a}$$

with $\mathcal{M}^s, \mathcal{M}^t, \mathcal{M}^a$ given by Eqs. (29). Because the DC part of the disturbances is also removed, this form of $\boldsymbol{B}^{c,s}$ does include implicitly the DC offset $\boldsymbol{G}^s$ introduced by the correction.

$\star$ We included a brief explanation on how to derive equations (29) after line 382[OM]/431-432[RM].

*3-7) page 17, lines 388-399, It is not clear what Gˆs expresses (it cannot be the absolute offset), and how equation (31) is derived.*

• Eq. (31) is the definition of $\boldsymbol{G}^s$. As the temporal average is taken over the entire time interval used to determine the correction matrices $\mathcal{M}$, $\boldsymbol{G}^s$ represents the difference between the mean values before correction and the mean values after the correction, i.e. a constant offset between the original measurements of the outboard sensor, $\boldsymbol{B}^{0,s}$, and the corrected field. To reduce the correction to a purely AC correction we must subtract this offset from from the corrected field, hence the correction which does not introduce a DC offset (defined as the difference between the mean values before and after the correction) is given by Eq. (28). We will revise the formulation in the manuscript to avoid the confusion between the corrected measurements which include the DC offset change due to the AC correction and $\boldsymbol{B}^{c,s}$ in Eq. (28) for which the DC offset change due to the AC correction is eliminated.

$\star$ We explained more clearly how the DC offset is changed by the proposed procedure and what the $\boldsymbol{G}^s$ vector represents. This involved changing equation (28) and adding the expression of the pure AC correction. The changes to the text were mostly after the line 389[OM]/439-445[RM].

*4) page 3 line 66, Because higher multipole moment ... Here authors say that they can ignore the contribution by higher degree moments. However, later they discuss under the assumption that one of the sensor pair is very closely located to the disturbance source, and therefore the contribution by higher degree moments cannot be negligible. Please make the descriptions consistent.*

• While in theory one could place a sensor so close to a disturber such that the octopole (or higher) contribution becomes significant, it is difficult to imagine a real life scenario where the dipole and quadrupole contributions of the disturber do not vastly overwhelm the octopole (or higher) contribution. We are indeed assuming in section 3.1 lines 151-153[OM]/175-176[RM] that the distance from one sensor to the disturber being currently cleaned is small *relative* to the distance to the other disturbers. This does not imply a distance small enough to make octopole and higher orders visible. It merely means that the disturbers are not equally distanced from the sensor in question and one disturber contribution dominates the others.

$\star$ On page 4 lines 107-110[OM]/126-127[RM] we state that the dipole and quadrupole contributions should not have comparable strengths at the sensor location. We changed this sentence to clarify that also higher multipoles should be weak compared with the dominant dipole or quadrupole being cleaned.

⋆ We clarified that even if we assume a small distance between one of the disturbance sources and one of the sensors, this does not mean that the higher order multipoles become significant (line 152[OM]/176-179[RM]).

*5) The proposed strategy to remove the noise argued in this paper seems to be inconsistent. In page 6, line 151, We now assume that one of the terms in Eq. (9) is much larger than the others. ... In page 10 line 2, the placement of the AMRs close to the disturbances sources. To do it, the authors should know the positions of the disturbance sources to locate the sensors nearby. It is inconsistent with the advantage of this method, allows the separation of disturbances generated by the spacecraft ... without prior knowledge about the positions of the disturbances sources. (page 5, line 134) Please make it consistent.*

• The observation is correct, of course some knowledge about the disturbers positions is necessary. For instance if a disturber is placed at equal distances from two sensors, the proposed procedure using those two sensors cannot work. It is also assumed that the boom-tip placed sensor is further away from disturbers than the other sensors, and that when the body-mounted sensors accommodations were decided at least some minimum information about the locations of major disturbers was available so sensors could be placed in their vicinity. However, apart from that, the positions of the disturbers do not enter in any way in the cleaning procedure, hence the statement on lines 134-135[OM]/156-168[RM].

⋆ We changed "positions" to "exact positions" on line 135[OM]/157[RM] and explained at the beginning of section 3.1 (after line 152[OM]/179-181[RM]) that even though the position of the disturbance sources does not enter the PiCoG formalism, some rough information on their location can help optimizing the sensor accommodation.

*6) page 9, line 232, 3-axis Flux Gate Magnetometer (FGM)... Is the outboard sensor built based on the design by Primdahl (1979) and inboard one is based on Acuna (2002) ? If not, please refer the papers more adequately.*

• Both sensors are neither designed similar to the sensors described in the early Acuña nor in the Primdahl papers. The references are for the fluxgate principle only. The Mario Acuña design consists of three single component sensors accommodated next to each other. The disadvantage of this approach is that the axes directions are determined by the ringcores and the pickup windings and thus they are not stabilised by the more robust feedback coils. In low field conditions (e.g. for the Voyager spacecraft) this is of no importance, however for e.g. MAGSAT it causes a significant uncertainty. Fritz Primdahl has therefore developed a vector compensated magnetometer in which a sensor similar to the sensors developed by Acuña has been placed in a sophisticated feedback coil system compensating the field for all three single sensors in all directions. Xavier Lalanne developed a very nice sensor in which 6 ringcores are placed on the sides of a cube inside a Helmholtz coil system. This is a great design because it is fully symmetric, however very elaborated. Our design is based on two crossed ringcores in the centre of a Helmholtz system. It is described in detail in the Themis Magnetometer paper by

Auster at al.

⋆ We replaced the references to the Acuña and Primdahl papers with a reference to Auster at al. paper (line 232[OM]/275[RM]).

*7)* *page 10, lines 263-268, I suppose that the sensing alignment relationship between the FGM and AMR sensors would significantly affect the result of the removal of the magnetic disturbances. Please describe the knowledge about the alignment relationship and its accuracy.*

● Actually the alignment between the FGM and AMR sensors plays no role in the PiCoG method. This is because the cleaning is performed only on the maximum variance components of the measurements which are independent on alignment.

*8)* *As for the March 4 case presented in this paper, magnetic disturbances are caused by multiple sources and they can be discriminated because the repetition periods are very different one another. The authors should discuss the condition regarding the repetition periods of the disturbances when the proposed method works well and when it does not.*

● That is correct. The proposed method works well when the polarization direction of the targeted disturbance is determined by the maximum variance direction. If the disturbances are in the same frequency range and therefore cannot be decoupled by using different window lengths one must either find their polarization direction using other means, or they must have magnitudes different enough such that the dominant disturbance – being currently cleaned – determines the maximum variance direction.

⋆ We included a new paragraph after line 361[OM]/406-410[RM] which discusses the importance of the characteristic time scale of the disturbances.

*9)* *The order of Figure 2 and Figure 3 should be changed since Figure 3 appears earlier in the text.*

● This is correct.

⋆ We changed the order of the figures 2 and 3

*10)* *The meaning of the word orthogonality in this paper is not clear. If it means linear independence, up to three independent, mutually orthogonal, simultaneously active disturbances can be separated using two sensors. (page 5, line 118) would not be correct. More than three disturbances may be separated if they are linearly independent. The statement in page 14, lines 323-332 should be revised.*

● In the manuscript, the word "orthogonal" has the common geometrical meaning: Two directions are called orthogonal if they form a right angle. We say that two disturbances are orthogonal if their maximum variance directions are orthogonal to each other. We will explain this better in the manuscript.

⋆ We made it clear that we mean orthogonality between the maximum variance directions (lines 118,123,322[OM]/138,141-142,366[RM]).

**11)** *Page 17, section 4.3, What is the advantage to remove the disturbances by the onboard processor ? Because it cannot be guaranteed that the coefficients do not change for long period, it would be much better to determine the coefficients from the raw data on the ground.*

• The coefficients were determined from raw data on the ground. Monitoring the magnetic field at geostationary orbit supplies important information about the space weather events reaching the Earth. On-board data cleaning provides near real-time accurate magnetic field data which is essential in this context. An added benefit is a four fold increase in the time resolution achieved by changing the telemetry from raw data from four sensors at one vector per second to cleaned data at four vectors per second.

⋆ we mentioned this in the Abstract and after line 55[OM]/65[RM]

**Referee #2 comments**

*... The presented method needs the disturbing sources to change with time (variance analysis). In spacecraft magnetical cleanliness DC magnetic disturbers play a big role. On the other hand the offset drift of fluxgate magnetometer is a known problem. Is the method valid for DC calibration? Otherwise write AC disturbance in line 5 (Abstract) and in line 496 (Summary and Conclusion). ...*

• If the mean field produced by a disturber is different from zero, i.e. there is a non-zero DC disturbance due to the targeted disturber – which is most of the times the case, the proposed method will automatically correct this DC disturbance if it depends in the same way on the distance to the source as the cleaned AC term (dipole/quadrupole). The total DC shift introduced by the final correction is contained in the $\boldsymbol{G}^s$ vector given by equation (31). However, even if the above condition is true, the method presented here does not provide the DC offset produced by completely time independent disturbers, therefore it cannot be used for the DC correction. Moreover, the internal offset drift of the sensors cannot be treated using the PiCoG technique.

⋆ We changed the lines in the Abstract and in the Conclusions according to the Referent's request. We also added one sentence stating that the proposed technique deals only with AC disturbances at the beginning of section 2 (line 66[OM]/76-77[RM]).

⋆ We revised the discussion in section 4.3 on the DC contribution introduced by the PiCoG correction.

*... The authors call their method to deal with the SOSMAG data PiCoG algorithm. It is more of a methodology than an algorithm that could be coded as is.*

• That is correct. We abused the word "algorithm".

⋆ We replaced "algorithm" with "technique/method" throughout the manuscript.

*In chapter 2.1 interesting formulas are deduced for dipole and quadruple fields. They are used to show, that the magnetic field of a low frequency source can be factorized in a time-dependant and a geometry part. But is not that clear anyway for quasi DC magnetic sources?*

• Indeed, equations (1) and (4)[OM]/(1),(5)[RM], which are just the expressions for the magnetic field produced by a dipole/quadrupole show the trivial fact that in this case the space dependence can be separated by the time dependence as a factor. The key relations here are equations (2) and (5)[OM]/(2),(6)[RM] which show that the magnetic field produced by a time-dependent dipole/quadrupole at a given location can be obtained using a *time independent* transformation of the magnetic field measured at a different location. This proves that it is in principle possible to use the measurements from one sensor to correct the measurements delivered by another sensor at different location. This is the foundation of our approach.

*Do the formulas for the geometry factors enter the evaluation?*

• No, the $\mathcal{T}$ matrices given by equations (3) and (6)[OM]/(3),(7)[RM] are not used as such by the PiCoG technique. They might be used perhaps in a very controlled environment when the positions of the sensors and disturbers, as well as the dependence on the distance to the source (dipole/quadrupole) of the disturber are precisely known. This work's goal is to provide a technique which does not require this information. However, the correction matrix $\mathcal{A}$ is related to the $\mathcal{T}$ matrix by the relation in line 146[OM]/170[RM]. This means that – once the $\mathcal{A}$ matrix is computed using the PiCoG method – one could derive the corresponding $\mathcal{T}$ matrix. This is beyond the scope of the present work.

*In chapter 3.1. it is assumed, that one of the magnetometers is very close to a disturber. Does the method also work, if that is not the case?*

• This depends on the specific sensors-disturbers configuration. E.g. if more disturbers of the same type occupy a volume which is small compared to the distance to the closest sensor, the method works even if the sensor is not closer to one of the disturbers than to the others. If the directions of maximum variance of two disturbers are orthogonal to each other and the disturbances have different time scales, again the method works even if the strength of the two disturbances are the same / disturbers placed at similar distances to the sensor. There are however situations in which the method does not work, as detailed in section 5.

⋆ We added a paragraph in section 5 after line 425[OM]/479-485[RM] which discusses the case of disturbances of similar strengths at one sensor location.

*Unfortunately none of modern methods for analysis of multivariate time series is used. Principal component analysis is one of them. It uses spectral analysis of the cross-covariance matrix of all additional measured magnetometer components with respect to the reference magnetometer. ...*

• We *do* use Principal Component Analysis (PCA) as a key method employed by PiCoG. We state this e.g. in lines 119-120, 164-166, 496-497[OM]/139-140, 202-203, 556-557[RM]. It is however not necessary to use all the measured magnetometer components at once as input for the PCA. In our case the PCA reduces to the determination of the variance principal system for the three components of the magnetic field. We indeed determine the direction of maximum variance from the eigenvectors of the co-variance matrix as described e.g. in section 1.4 of *Time Series Data Analyses in Space Physics, Song and Russell, SSR (1999)* or in *Analysis methods for multi-spacecraft data, Sonnerup and Scheible, ISSI Sci. Rep. SR-001, p185-220, Ed Paschmann and Daly, (1998)*. We do not perform a spectral analysis though because it was not necessary for our specific problem. Of course, if one knows – or determines – in advance that the disturbance to be removed is confined in a specific frequency band, one may perform the PCA in the frequency domain and select the eigenvectors corresponding to the frequency band of the disturbance.

⋆ We added the above two references to the text (line 165[OM]/194[RM]) and we mentioned the possibility to use band pass filtering after line 425[OM]/481[RM].

*... The results of sec. 2.1, that means the known geometry factors, add information not been used in standard methods. Using for example the geometry factors for better identifying disturbing time series in the data, or for better determining the distribution of disturbing signal to the different magnetometers would render this paper interesting to a broader public.*

• It is true that in this work we did not directly exploit equations (3) and (6)[OM]/(3),(7)[RM] which give the exact expression of the "propagator" matrix $\mathcal{T}$ which allows computing the disturbance at one point in space once the disturbance at another point is known. This would allow cleaning the disturbances using a precise model representing the disturbance sources and the sensors positions. However, this is not the goal of the present work. Here we determine the correction matrices $\mathcal{A}$ – which are equivalent with the $\mathcal{T}$ matrices – solely from the available measurements. Of course, it might be possible to develop an entirely different method using the $\mathcal{T}$ matrices computed based on the precise positions of the disturbers and of the sensors. However, making more intensive use of the $\mathcal{T}$ matrices is not necessary in the context of the present work.

⋆ We mentioned the possibility of exploiting equations (3) and (6) after line 133[OM]/154-156[RM].

*The time dependence of the disturbing signal is not at all used in chapter 3, where the PiCoG Algorithme is defined.*

• The time dependence is implicitly used through the fact that the correction is applied to the principal variance component. Also the scaling factor $\alpha$ defined in section 3.1, equation (14) is determined using the variance of the measurements, therefore using the time dependence of the disturbing signal.

*... Nevertheless during the actual data evaluation the authors implicitly use the time dependence by looking at different time periods, with different sources active. Fig. 2 shows the distribution of directions on a sliding window. Later on, in chapter 4. ramps and spikes are used to validate the result. These tricks should be included in the PiCoG algorithm.*

• We made efforts to keep the PiCoG method described in section 3 as general as possible (please see also Referent #1 comment 2). Including procedures specific to our particular application of the method to the SOSMAG data would in our opinion induce confusion to the reader. Presenting these procedures in the application section on the other hand, lets the reader decide for him/herself if these procedures are appropriate or not for his/her problem at hand. Even for our specific problem we did not used the same procedures from the beginning to the end: For the AMR correction we determined the maximum variance direction using just one step-like disturbance, while for the FGM-FGM correction we decided for a statistical approach using a sliding window.

*The variance of measured date is used. That means field sources (ambient and disturbing*

*fields) are understood as random processes. The motivation is not clear.*

• It is true that variance normally refers to the the deviation of a random variable from its mean value. A certain randomness is introduced by the ambient field. However, in the context of the present work, the random/non-random character of the disturbance plays no role. We use the variance only as a measure of how strong the AC disturbance is in each direction, and through PCA we determine the direction in which the variance is largest therefore the disturbance is strongest.

⋆ We explained better how the variance analysis is used by PiCoG and clarified that for our purposes it is not necessary for the disturbance or for the ambient field to be generated by a random process (after line 164[OM]/196-197[RM]).

*...The step amplitudes in all components could directly be used to deduce the geometry factors (Component of Matrix A in formula 10) between different magnetometers.*

• The correction matrix $\mathcal{A}$ is composed from a rotation and a scaling. We don't see a direct way to deduce the $\mathcal{A}$ matrix from step amplitudes. After the rotation in the VPS one could indeed determine the amplitude of the steps as we did in section 5 and from them derive the scaling factor $\alpha$ in equation (13a). We think however that equation (14) gives a more general solution. Both estimates of the $\alpha$ factor are susceptible to improvements anyway as mentioned in lines 273-274[OM]/316-318[RM].

*Fig. 4. Shows magnetometer values in the coordinate system orientated along the main axes of the data variances ellipsoid (VPS system). The figure shows, that the spike signal is still present in the z- and in the y-direction. Accordingly the VPS x-direction does not point along the spike disturbance.*

• This is correct. The $x$-axis of the VPS in Figure 4 is aligned with the variance direction of the highest frequency disturbance (first to be cleaned), distinct from the direction of the spikes. This is discussed on lines 321-324[OM]/365-368[RM]. The VPS in which the data in Figure 5 is represented has its $x$-axis aligned with the direction of the spikes.

*L39: The PiCoG Process also is not running on the SC.*

• The PiCoG technique delivers the correction matrices $\mathcal{M}$ which are uploaded to the spacecraft and used for onboard data cleaning. As far as we understand, the *Poppe et al. (2011)* procedure cannot be reduced to a simple linear combination which can easily be implemented onboard.

*L48: This is the case if only variances are looked at. But the authors look later at ramps and spikes. They can even be identified, if they point along the ambient field.*

• We use the PCA/variance analysis also for the ramps and for the spikes. This is specified in multiple places in section 4.2. One could use other methods to determine the direction of the ramps or spikes disturbances (even manually perhaps), but the PCA delivers the correct directions and the scale factors in an automatic fashion.

⋆ We emphasize in the text the use of PCA for the MD disturber: line 269[OM]/312[RM]

*L48: The term principal component is misleading. It usually refers to direction of a main axe of the stray ellipse in a multivariate random process.*

• The term "principal component" in the text does indeed refer to the main axis of the variance ellipsoid. The fact that the disturbance is not a random process does not affect neither the application of the PCA nor its results.

⋆ We mentioned on line 165[OM]/196-197[RM] that we do variance analysis without implying random processes.

*L107: Perhaps better: a collection of dipoles will in general generate multipole moments*

• The suggested formulation is indeed better. Thank you.

⋆ We changed the text according to the Referent's suggestion.

*L 151: Which term is much larger? Would a strong disturber really make only one term large?*

• Since the summation index $q$ in equation (9) refers to the disturbance sources – as detailed in line 137[OM]/160[RM], the term corresponding to the strong/close disturber will be larger. A strong disturber will only affect the corresponding term in the sum.

*L137 this sentence would be more readable, if the summation symbol was omitted.*

⋆ We re-phrased this sentence to make it more readable.

*L157: Do you mean: disturbing magnetic moments are fixed in direction with moments changing with time?*

• Yes, this is what we mean.

⋆ We adapted the text.

*L158: The stray field of one disturber has a constant direction in the magnetometer system. No need for a new coordinate system.*

• The stray field of one disturber has indeed a constant direction in the magnetometer system. However, a new coordinate system is needed to align this direction with one of the coordinate system axes (in our case the $x$-axis).

*L166: Using this VPS suggests, that the disturber itself is a multivariate random process. But that is not the case. The VPS-x direction can be calculated by correlating the disturbing field strength with the measured x-, y- ,z- components. The term variance principle system is misleading. The reader could get the impression, that principal component analysis was done.*

• PCA was in fact done in order to obtain the disturbance direction. As stated before, the non-random character of the disturbing field is not relevant in this context. One could probably obtain the disturbance direction by minimizing the correlation between the disturbing field strength and the measured $y$ and $z$ components using as free parameters

the angles of rotation for the new system. We do not see the advantage in using this alternative method.

$\star$ We mentioned on line 165[OM]/196-197[RM] that we perform variance analysis without implying random processes.

*L167: Are the alpha i,j in Eq. 13a the same as the A i,j in Eq. 10? Than please use the same denomination.*

• They are not the same. $\mathcal{A}^{ij}$ in equation (10) is the matrix used to correct sensor $i$ measurements using sensor $j$ measurements. $\alpha^{0,ij}$ in equation (13) is a scalar scaling factor given by equation (14) for the first order correction.

*L191: It is not clear to me if the bs are known at this point and if yes, where they are calculated.*

• The disturbance $b$ at the sensor position is not known at this point. We only make use of the dependence on the distance of $b$ as stated in lines 193-195[OM]/235-237[RM].

*L211: Do you mean if stray fields of different disturbers are not coincident at the magnetometer location?*

• We mean "if stray fields of different disturbers do not share the same direction at the magnetometer location".

$\star$ We have revised the text.

*I quit following the text here because the authors use a matrix notation, where I guess vectors of stray fields are sufficient.*

• We do not see how to concisely write the relations without using matrix notation.

*Conclusion: The paper is an excellent report on how the authors achieved to clean and calibrate SOSMAG data. However the term principal component technique in the title is misleading. The authors should revise the method and try to use or at least refer to standard methods for multivariate data analysis and, if possible, expand them to produce a paper of more general interest.*

• We thank the Reviewer for the appreciative comment. However, as explained above on several occasions, PiCoG *is* using principal component analysis as a essential tool, therefore we believe the title is appropriate. We changed the title nevertheless, please see our answer to the Reviewer's reaction bellow.

$\star$ We change the title to "Maximum Variance Gradiometer technique for removal of spacecraft-generated disturbances from magnetic field data".

**Referee #2 reaction to response**

**"PiCoG algorithm"** *: Would you think that the following procedure is in line with what you did. Could this be a line out of an algorithm?*

*1. Define one of the N instruments as "reference instrument".*

*2. Calculate the differences of all instruments and the reference instrument.*

*3. In all difference signals, identify the instrument j showing the strongest disturbance.*

*4. Use this difference to correct all instrument readings using Eg. 13*

*5. For the next iteration start over with 2. disregarding instrument j.*

- No, we use a different procedure:

Assume three magnetometers, $m_0$, $m_1$ and $m_2$ and two disturbance sources, $d_1$ and $d_2$. Assume we define instrument $m_0$ as reference instrument. Assume the dominant disturbance at the instrument $m_1$ location comes from the source $d_1$ and the dominant disturbance at the instrument $m_2$ location comes from the source $d_2$.

Assume the difference is largest for the instrument $m_1$, i.e. $|\text{var}(\Delta \boldsymbol{B}^{01})| > |\text{var}(\Delta \boldsymbol{B}^{02})|$. In these conditions equations (13) will work correctly to clean the disturbance $d_1$ from the measurements taken by the reference instrument $m_0$ and – if desired – also from the measurements taken by the instrument $m_1$.

However, as explained in lines 170-173[OM]/207-209[RM], beside the $(\Delta \boldsymbol{B})_x$ term, which is written in the VPS of the difference $\Delta \boldsymbol{B}$, all other terms in equations (13) are written in the VPS of the measurements at the respective instruments. The maximum variance $x$-axis computed for the instrument $m_2$ will be aligned with the direction of the disturbance $d_2$ at the location of the instrument $m_2$ which in general will be different from the direction of the disturbance $d_1$ at the location of the instrument $m_2$. Therefore equations (13) used as suggested by point 4 above, would apply the correction for the disturbance $d_1$ to the wrong component of the measurements taken by the instrument $m_2$. Moreover, the scaling factor computed using the variance of the measurements from the instrument $m_2$ will also be wrong.

In contrast, the PiCoG technique uses one instrument pair at a time: After points 1-3 above, we clean the (strongest) disturbance $d_1$ from the measurements of the reference instrument $m_0$. Afterwards we compute the difference between the cleaned measurements from the reference instrument $m_0$ and the measurements from the instrument $m_2$. This difference will now reflect the disturbance $d_2$. We determine the VPS of the difference and of the measurements from the instrument $m_2$ and we finally apply again equations (13) to clean the disturbance $d_2$ from the cleaned measurements taken by the reference instrument $m_0$.

Note that the method works without knowledge about the exact positions of the sources and – after the correction matrices are determined on ground – it works with simple multiplications and additions, which can be done in real time by the onboard software.

**PCA:** *You determine the main axes in the 3D distribution of magnetometer measurements and in the distribution of differences between two different magnetometers. Then you assume that your $\alpha$ can be calculated based on the quotient of variances along these*

*main axes (Eq. 14). This is a very bold assumption and you named quite some requirement for this assumption.*

• Up to a constant factor, the difference $\Delta \boldsymbol{B}$ is the same as the disturbance at the sensor to be cleaned (same time dependence). The factor is the ratio between the amplitude of the difference and the amplitude of the disturbance at the sensor to be cleaned. This can be directly derived from the variances. As mentioned on line 174[OM]/215-216[RM], equation (14) gives a first order estimation of the scaling factor $\alpha$. This estimation may deviate from the exact scaling factor due to large ambient field fluctuations or due to additional disturbances with the same polarization direction as the disturbance to be cleaned. To improve this value one may for instance minimize the correlation between the corrected measurements and the disturbance represented by the difference $\Delta \boldsymbol{B}$ as we note on lines 271-274[OM]/315-318[RM]. However, in our case this proved not to be necessary.

⋆ We explain this better after line 173[OM]/213-216[RM].

*Asking for PCA, I meant to use PCA in the 3N dimensions of all available measured time series. If PCA is referred to in the title the reader will expect it to be used on the multivariate time series*
$(X_1(t), Y_1(t), Z_1(t), \Delta X_{21}(t), \Delta Y_{21}(t), \Delta Z_{21}(t), \Delta X_{31}(t), \Delta Y_{31}(t), \Delta Z_{31}(t), ...).$
*"1" being the reference magnetometer. This automatically produces what you call VPS-x directions (as components of the largest eigenvector).*

• PCA done in 3 dimensions is not something unusual in space physics data analysis, see e.g. section 1.4 "Principal Axis Analysis" of *Time Series Data Analyses in Space Physics, Song and Russell, SSR (1999)*. There might be a way to use PCA in $3N$ dimensions for cleaning multi-sensor data, but the exact implementation of this is not obvious to us. The maximum eigenvalue and the corresponding eigenvector derived for the multivariate time series suggested by the Reviewer would somehow mix the reference instrument measurements with the differences between those measurements and the measurements from all other instruments. Moreover, as explained above, if different disturbances affect different sensors, they will also be mixed together, even if initially they were decoupled from one another. This is exactly what we are trying to avoid. Even if perhaps possible, at the moment we do not see how a technique based on PCA in $3N$ dimensions can be implemented for cleaning multi-sensor data. As we showed by applying it to SOSMAG data, the (3D PCA) procedure proposed by us works well to decouple and clean spacecraft disturbances, and, in our opinion, is general enough to be easily adapted to other multi-sensor configurations.

⋆ We explained after line 159[OM]/194-196[RM] that we determine the principal components using only the 3D time series from individual sensors.

*... Please judge for yourself whether the reference to PCA in the title is really justified.*

• We realise that some readers might expect a treatment along the lines suggested by the Reviewer, therefore we change the title to "Maximum Variance Gradiometer technique

for removal of spacecraft-generated disturbances from magnetic field data".

*But even PCA and factor analysis do not deliver unique results. In PCA geometry factors are completely ignored. Therefore exploitation of Eg. 3 and Eq. 6 would introduce a completely new idea going further than what can be done by PCA.*

• Using the equations (3) and (6) would be indeed a very different approach from the one presented by us. It is definitely worth exploring ways to use these relations to develop new methods – perhaps model-based – for cleaning multi-sensor data. Once developed, such methods could be combined with the PiCoG technique to improve the results, but this should be the focus of another study.

*On page 6 between line 157 and line 180 you argue very intuitively. This lack of mathematical rigor should be mitigated using PCA in the way I proposed.*

• As explained above, a direct application of PCA in $3N$ dimensions is not a solution for our problem. On page 6 we write down the expressions of the corrected measurements under the stated assumptions. We do not see where the lack of mathematical rigour lies.

*It is absolutely not clear how Eq. 10 follows from Eq. 9. I even doubt, that a linear relation between the correction value for $B^i$ and the $\Delta B^{i,j}$ exists. This is only true if only one single disturber is on. I guess Eq.10 is the first order approach assuming that a certain disturber is very prominent (at a certain time span) in the difference $\Delta B^{i,j}$. Please clarify and explain that in the text.*

• Equation (10) does indeed not follow equation (9) in the general multiple disturber case. We will reformulate the text to better explain that equation (10) is valid for single disturber case.

⋆ We changed the text after line 143[OM]/166-168[RM] to better explain that equation (10) is valid for single disturber case.

[revised manuscript text omitted]